# Trial-Level and Contiguous Syntactic Adaptation: A Common Domain-General Mechanism at Play?

Varvara Kuz *, Fangzhou Cai, Keyue Chen, Jiaxin Chen, Xuzi Qi, Clement Veall, Yuanqi Zheng, Zhengping Xu and Andrea Santi

Department of Linguistics, Division of Psychology and Language Sciences, University College London, London WC1N 1PF, UK; fangzhou.cai.21@alumni.ucl.ac.uk (F.C.); keyue.chen.19@ucl.ac.uk (K.C.); jiaxin.chen.20@alumni.ucl.ac.uk (J.C.); xuzi.qi.21@alumni.ucl.ac.uk (X.Q.); c.veall@ucl.ac.uk (C.V.); yanqi.zheng.20@alumni.ucl.ac.uk (Y.Z.); zhengping.xu.19@alumni.ucl.ac.uk (Z.X.); a.santi@ucl.ac.uk (A.S.)
* Correspondence: varvara.kuz.19@ucl.ac.uk

**Abstract:** Garden-path sentences generate processing difficulty due to a more preferred parse conflicting with incoming parsing information. A domain-general cognitive control mechanism has been argued to help identify and resolve these parsing conflicts. This cognitive control mechanism has been argued to underlie adaptation to garden path processing at the trial level (conflict adaptation) and contiguously over the experiment (syntactic adaptation) in independent literature. The strongest evidence for its domain generality comes from garden-path processing being facilitated when preceded by a non-syntactic conflict (e.g., Stroop). This has been reliably observed in the visual world paradigm, which, like Stroop, requires irrelevant visual information to be suppressed. We tested the domain generality of conflict adaptation and its relationship to contiguous syntactic adaptation across four experiments (*n* = 562). To eliminate the visual object confound, the Stroop task was followed by a sentence-reading task. We observed Stroop and ambiguity effects, but no conflict adaptation in each experiment. Contiguous syntactic adaptation was replicated and most compatible with the parser changing its expectations and/or improving revision. While the data largely fail to support a domain-general cognitive control mechanism, a language-specific one could operate in both trial and contiguous syntactic adaptation and is worth future exploration.

**Keywords:** syntactic adaptation; conflict adaptation; cognitive control; sentence processing

## 1. Introduction

It seems irrefutable that once a language is acquired to proficiency it remains adaptable to experience within a local environment. Most of us can think of particular locations, social contexts, groups of friends or even a single friend that our language comprehension/production adapted to either in terms of accent, wording, or phrasing. For example, for someone moving from North America to Britain, learning that the term 'tea time' can refer to a meal that does not even involve any tea is surprising at first, but can gradually become the norm in comprehension and production. Or the adoption of a friend's "y'all" to avoid the ambiguity of plural vs. singular "you" in one's own dialect of English. While these intuitive examples consider different dialects of English, adaptation has also been observed within a dialect and, in particular, in the resolution of syntactic ambiguities. For example, you might find it easier to read the newspaper headline, "EU Settlement Scheme Delays Leave People 'Unable to Get Jobs or Housing'"(Bulman 2020), after having read "Missing woman remains found"(Liberman 2015). In both cases, there is an ambiguity as to whether a word (i.e., *delays, remains*) is acting as a verb or a noun and our preference to interpret it as a verb causes a boggle at the actual verb (i.e., *leave, found*) and reinterpretation occurs. This is standardly referred to as a garden path structure and it has been found that participants' processing of garden paths is facilitated with increased exposure (Fine and Jaeger 2016; Fine et al. 2013; Yan and Jaeger 2020). What is heavily debated are the

mechanisms that allow for syntactic adaptation, at what timescales they operate, and the degree to which they operate solely within the parser or are dependent on the support of domain-general mechanisms. These questions are the focus of the current study.

Investigations into these questions have often focused on the difficulty of processing garden-path sentences compared to unambiguous versions of the final correct interpretation. For example, in sentence 1a. *unloaded* is ambiguous between a reduced relative clause (RRC) and a main clause (MC) verb. Initially, it will be interpreted as the more probable MC verb, but this interpretation becomes problematic once the parser encounters the actual main verb *found*, which causes *unloaded* to be reanalysed as the past participle within a relative clause (RC). This can be phrased unambiguously, as in 1b., by introducing the RC with *who were*, so *found* can be immediately interpreted as the past participle within the RC. Studies have shown slower reading times at the disambiguating region (i.e., *found*) in the ambiguous compared to the unambiguous sentence (Fine and Jaeger 2016; Fine et al. 2013; MacDonald et al. 1992; Yan and Jaeger 2020). This *ambiguity effect* is often interpreted as a consequence of syntactic *surprisal* (the degree a word is unexpected given the syntactic context) and/or subsequent revision.

1a. The alert soldiers unloaded from their helicopter found their target quickly.

1b. The alert soldiers who were unloaded from their helicopter found their target quickly.

However, it has also been argued that the resolution of a parsing conflict, as occurs in sentence 1a., could be assisted by a domain-general cognitive control mechanism (Novick et al. 2005, 2014). Cognitive control allows us to switch from an automatic to a controlled behaviour in response to changing goal demands, requiring mental flexibility and the manipulation of information available at the moment (Cocchi et al. 2013). The conflict monitoring account of cognitive control assumes mechanisms that monitor for conflict and then help to resolve it (Botvinick et al. 2001). As such, they may help us resolve the conflict between the initial misparse that would be automatically projected, and later incoming information incompatible with it to reach the final goal—correct interpretation. As this example illustrates, garden-path structures provide a rich playing ground to look at what can be learned: the probability of each parse in a temporarily ambiguous structure, reanalysis procedures, or higher-level oversight by a domain-general mechanism for identifying and resolving conflicting representations.

## 1.1. Two Related Adaptation Results Explained under a Domain-General Cognitive Control

Two adaptation phenomena that operate over different timescales have provided arguments for domain-general cognitive control mechanisms assisting with improved garden path processing: *syntactic adaptation* and *conflict adaptation*. Syntactic adaptation refers to the observation of a progressively decreasing ambiguity effect over the course of weeks (Wells et al. 2009) or even an experiment (Fine and Jaeger 2016; Fine et al. 2013). Conflict adaptation generally refers to the facilitation of resolving conflicting representations (like in garden paths) through exposure to an immediately preceding trial/stimulus that also contains conflicting representations compared to low-conflict or non-conflicting representations. Interestingly, studies have shown conflict adaptation can occur across different forms of representational conflict, for example from a non-syntactic conflict (Stroop) to a syntactic conflict (e.g., garden-path comprehension) (Hsu et al. 2021; Hsu and Novick 2016; Kan et al. 2013). According to *conflict monitoring theory* (Botvinick et al. 2001), detecting conflict during a high-conflict trial engages cognitive control that overrides the automatic behaviour for the goal-directed task and then remains actively engaged long enough to assist in identifying and/or resolving conflict for a short period immediately after.

Both conflict adaptation and syntactic adaptation describe adaptation, or facilitation, to the disambiguation of competing parses, but on different timescales. In syntactic adaptation, experience is accumulated over the course of an experiment, and as such it is *contiguous*; conflict adaptation on the other hand depends on the activation of (a) mechanism(s) encountered in the immediately preceding trial or task, that is thought to dissipate with time, on the scale of a trial, and can therefore be seen as a *trial-level* adaptation. These

two results have been collectively interpreted by some (Hsu and Novick 2016; Thothathiri et al. 2018; Yan and Jaeger 2020) as the parser working in tandem with a domain-general cognitive control mechanism that (partially) underlies both of these phenomena. The role of domain-general mechanisms in language comprehension is a controversial topic that is debated across a variety of fields—psycholinguistic and neurolinguistic—with evidence presented from both sides (Fedorenko 2014; Hsu et al. 2017; Hsu and Novick 2016; Kan et al. 2013; Novick et al. 2014; Novick et al. 2005; Patra et al. 2023; Vuong and Martin 2014). Resolving this debate will require a thorough analysis of the methods and designs that provide evidence in one way or another. In the following sections, we will review past work on both contiguous- and trial-level adaptation to garden-path structures and the evidence for the involvement of cognitive control in both of these phenomena. We will then identify some concerns for this work before presenting four experiments to further explore contiguous and trial-level adaptation within the same dataset. The particular goal in doing this is to further assess the evidence in favour of a conflict-based domain-general cognitive control mechanism in facilitating syntactic ambiguity resolution over time vs. the alternative accounts.

*1.2. Garden-Path Facilitation: Training on Sentences over Weeks*

Some of the earliest evidence showing that we can become more efficient at processing or avoiding these syntactic conflicts came from a training study. Wells et al. (2009) studied object (2a) and subject relative (2b) clauses that, like garden-path sentences, give rise to a temporary ambiguity. At the complementiser *that*, there is ambiguity as to whether the noun phrase *the reporter* is the subject or object of the RC. Given the subject relative is more frequent and easier to process, the parser is guided to expect a subject relative. However, conflict (aka high surprisal) arises in the object relative in 2a. when the parser encounters the subject, *the senator* (Hale 2001; Levy 2008). Further, in completing the dependency between the reporter and its predicate *attacked*, conflict may arise in retrieving the correct object, as there is an intervening noun phrase, *the senator*, that matches many of its features (i.e., it is animate, definite, and nominal) (Van Dyke and McElree 2006, 2011). These two possible sources of conflict have been argued to increase processing demands in the object relative over the subject relative. Participants in Wells et al. (2009) saw an equal number of sentences with object RCs (like 2a.) and subject RCs (like 2b.) over a period of 4 weeks and were tested on their reading times pre and post training. They found an asymmetric learning effect, where the decrease in reading times for object relatives at the main verb (i.e., *admitted*) was greater than that of subject relatives during the post-test phase. A control group saw alternative difficult structures during training and did not demonstrate this improvement. This shows that the adaptation was specific to the structure and not just general sentence difficulty training. This demonstrates that with higher exposure to the less frequent structure, it can be processed more easily.

2a. The reporter that the senator attacked admitted the error.
2b. The reporter that attacked the senator admitted the error.

However, Wells et al.'s (2009) results could also be compatible with cognitive control training over those weeks. The control group were trained on sentences (complement clauses, coordination, and other 'difficult' sentences without description) that do not seem to involve ambiguity or stated otherwise 'conflicting' parses. Thus, this training may not have recruited cognitive control (to the same degree as for the experimental group), which would explain the lack of a post-training improvement. Support for this argument comes from early results showing the involvement of domain-general cognitive control in syntactic ambiguity resolution that likewise used a training paradigm (Hussey et al. 2017; Novick et al. 2014). In Novick et al. (2014), participants were trained on a series of non-linguistic cognitive tasks targeting a variety of cognitive functions over a period of three to six weeks. This included conflict resolution training with an n-back task with lures, where participants had to indicate whether the currently displayed letter appeared 'n' (where $n = 1, 2, 3$) items prior. Conflict was achieved by using 'lures' or target letters presented in

a position before or after the targeted *n*th-back position; the conflict between the identity of the letter and its position is theorised to recruit the conflict resolution component of cognitive control (Burgess et al. 2011; Gray et al. 2003; Kane et al. 2007), which is also thought to help recovery from syntactic conflict in garden-path sentences. Participants' garden-path processing was tested before and after training on these various tasks with sentences containing the subject–object ambiguity. In 3a., the NP *the jewellery* is initially interpreted as the object of *hid* due to the parser's preference for late closure but is then reinterpreted as the subject of *sparkled*, requiring early closure of the subordinate clause. This ambiguity is avoided with the reversal of the two clauses in 3b.:

3a.  While the thief hid the jewellery that was elegant and expensive sparkled brightly.

3b.  The jewellery that was elegant and expensive sparkled brightly while the thief hid.

They found that an improvement in ambiguous sentence comprehension at post-test was only observed when the participant demonstrated improved performance on the n-back task over the training period. Improvement in ambiguous sentence comprehension was not found with improvement in the other cognitive tasks that did not involve conflict or in those participants that did not show improvement in the n-back task. These results suggest a cognitive control mechanism is recruited by both types of tasks to allow improvement in the training task to transfer to sentence processing.

However, the design of Novick et al. (2014) did not include a low-conflict n-back task, limiting conclusions about the specific mechanism that contributed to the improvement in syntactic parsing performance. Hussey et al. (2017) addressed this by having training groups that were either exposed to the n-back with lures or n-back alone. The effect of cognitive control training on garden-path processing was assessed using the same subject–object ambiguity sentences as in Novick et al. (2014), as well as subject and object RC sentences like those in Wells et al. (2009). Hussey et al. (2017) argue that the object RC sentences do not involve conflict, in contrast to what we argue above based on others' discussion of its ambiguity and/or interference in memory (Futrell et al. 2020; Traxler et al. 2002; Traxler et al. 2005; Wells et al. 2009). Eye-tracking while reading was used to assess the processing of both sentence types pre and post training. While the group trained on the n-back with lures task demonstrated a significant reduction in regression-path reading times at the disambiguating region in the ambiguous subject–object sentences, it was not significantly different from those trained on the simpler n-back task. This key interaction, however, did appear in second-pass reading times at earlier regions, which were reread faster by high-conflict trainees. Thus, n-back training (but not specifically on the high-conflict version) seems to have a benefit at disambiguation, while training on the conflict monitoring version of the task results in some unique improvement in terms of resolving the conflict after it is detected. Interestingly, they did not find that training on the n-back with lures task had any effect on object relative processing across their critical region (the entire RC). As it is a null effect, it is hard to make conclusions, especially since the regions analysed by Hussey et al. (2017) and Wells et al. (2009) differ, but it leaves open the possibility that Wells et al.'s (2009) training effect was unrelated to cognitive control improvement.

*1.3. Garden-Path Facilitation: Training on Sentences over ~1 h Experiment*

More recent data suggest that adaptation to syntactic processing can be achieved on a much shorter time scale—over the course of a typical psycholinguistic experiment (less than an hour). Fine et al. (2013) exploited the same RC/MC ambiguity as in 1a.-1b. and found a gradual decrease in the ambiguity effect as participants were exposed to incrementally more RC structures over the course of an experiment, an effect that has come to be known as *syntactic adaptation*. Syntactic adaptation has been replicated in further studies with the same temporary ambiguity as well as others (Atkinson 2016; Fine and Jaeger 2016; Kaan et al. 2018; Prasad and Linzen 2021; Yan and Jaeger 2020). Expanding on Wells et al.'s (2009) interpretations, Fine et al. (2013) interpreted this adaptation as a consequence of the parser changing distributional parsing probabilities. However, a

likely requirement for this to occur is that there be a dramatic change in the probability of exposure to the less preferred parse in the experimental environment compared to the natural environment. While exposure to the RC and MC parse was equal (probability of 0.50) in the experiment, a corpora analysis indicated the RC probability was only 0.008 while the MC probability was 0.70. This change in the probability of occurrence was argued to lead the parser to progressively expect the RC to parse more over the course of the experiment. This interpretation allows improvement in garden-path processing to occur within the parser.

Alternatively, others have argued that the observed adaptation may actually reflect a simple learning of the artificial self-paced reading task asymmetrically affected by sentence difficulty (i.e., RCs take longer to process from the outset and therefore can show a larger adaptation effect) rather than a change in the probabilities associated with different parses (i.e., it is not important that it is an RC but that it is difficult to comprehend; (Prasad and Linzen 2021)). Yan and Jaeger (2020) brought the same sentences and design as Fine et al. (2013) into an eye-tracking-while-reading study to test whether the effect was observable outside of self-paced reading. They again found evidence for syntactic adaptation, arguing against the idea that it is simply about learning the novel task of self-paced reading and its interaction with difficulty. However, the effect was observed in a measure that is typically interpreted as corresponding to reanalysis processes rather than syntactic surprisal. This led the authors to suggest that the syntactic adaptation effect may stem from more efficient reanalysis rather than a change in expectation to the RC parse. This could either occur within the parser or be supported by a domain-general cognitive control mechanism becoming more efficient as the same reanalysis was replicated over time, for example, as an attentional and/or inhibitory mechanism was repeatedly paired with the incorrect parse. This represents a form of training of these mechanisms similar to the work of Novick et al. (2014) and Hussey et al. (2017) but over the course of an experiment rather than weeks. Collectively, the works looking at syntactic adaptation with training either over weeks or within an experiment argue for the effect being structurally specific, or at least dependent on conflicting parses, rather than to general difficulty, but there remains the question as to whether the learning is occurring solely within the parser or is also dependent on domain-general cognitive control mechanisms.

### 1.4. Garden-Path Facilitation over Trials

To test for cognitive control and syntactic ambiguity resolution having a causal effect on one another dynamically, Kan et al. (2013) exploited the phenomenon of conflict adaptation. Conflict adaptation was initially observed with the Stroop task (Gratton et al. 1992). In the Stroop task, the participant is to identify the font colour of a colour word. In the congruent or low-conflict condition, the font colour is the same as the colour word (e.g., RED written in red). In the incongruent or high-conflict condition, the font colour is different from the colour word (BLUE written in red). It is a well-documented finding that congruent trials take less time to complete than incongruent trials (MacLeod 1991), and this difference in reaction time is known as the *Stroop effect*. Conflict adaptation is the finding that the reaction time for an incongruent trial is lower if it directly follows another incongruent trial than a congruent one (Gratton et al. 1992). Conflict monitoring theory (Botvinick et al. 2001) proposes that the conflict between the two informational sources—the colour of the ink and the meaning of the word—during incongruent trials is resolved through cognitive control that helps inhibit the automatic but irrelevant action of word reading and instead focus on the relevant dimension of font colour. Cognitive control remains activated and helps resolve conflict on a subsequent incongruent trial faster, resulting in trial-to-trial 'conflict adaptation'. Thus, conflict adaptation describes how we utilise cognitive control to adjust our behaviour online in response to varying task demands. Kan et al. (2013) applied this in an interleaved design where the participants would first read a temporarily ambiguous or unambiguous sentence (4a.-4b.) in a self-paced reading manner and then complete a congruent or incongruent Stroop trial. In ambiguous sentences like 4a., the

interpretation of the NP *the contract* as the direct object of the verb *accepted* would become unacceptable at the disambiguating region *would have* where it turns out to be the subject of the complement clause.

4a.  The basketball player accepted the contract would have to be negotiated.

4b.  The basketball player accepted that the contract would have to be negotiated.

They found that the incongruent Stroop condition demonstrated faster reaction times and higher accuracy when following an ambiguous sentence condition compared to an unambiguous sentence condition. The performance on the congruent Stroop condition was unaffected by the sentence condition it followed. This was interpreted as the high-conflict trial activating conflict-monitoring mechanisms and remaining active long enough to help resolve conflict in the subsequent high-conflict trial. The fact this occurred despite the conflicts being qualitatively different between the two trials demonstrated a causal effect of a domain-general conflict monitoring mechanism.

However, a potential concern with this design is that it does not allow for any comprehension questions to follow the critical sentences, thus it is not known whether the participants achieved the correct interpretation. More concerning is that the findings were not fully replicated in two recent studies[1], despite having more highly powered designs and replicating the standard Stroop and ambiguity effects. Aczel et al. (2021) used 132 participants vs. 41 used in Kan et al. (2013), and only replicated the accuracy but not the RT results. Dudschig (2022) used 93 participants and found a reverse conflict adaptation effect, with the Stroop effect in RT becoming larger after the ambiguous sentences; she found no significant effect for Stroop accuracy.

Hsu and Novick (2016) followed a similar idea as Kan et al. (2013) but with two significant modifications: (1) the task order was reversed (Stroop—Sentence Comprehension), and (2) the self-paced reading was substituted with the visual world paradigm. As such, the participants first completed an incongruent or congruent Stroop trial after which they saw a visual display and heard an ambiguous or unambiguous sentence. Ambiguous sentences like 5a. used a PP that was temporarily ambiguous between being a goal vs. modifier of the object. The PP is initially preferentially interpreted as the goal location of the object, *the frog*, but when hearing the disambiguating second PP, *into the box*, it becomes clear that the first PP is in fact a locational modifier. In 5b., *that's* unambiguously identifies a RC or modifying structure. After listening to the sentence, the participants moved the target object to the correct goal.

5a.  Put the frog on the napkin into the box.

5b.  Put the frog that's on the napkin into the box.

Hsu and Novick (2016) found that when the ambiguous sentence was preceded by an incongruent Stroop, the participants looked more to the correct goal and committed fewer action errors than when it was preceded by a congruent Stroop. There was no difference in performance in the unambiguous sentence trial whether preceded by an incongruent or congruent Stroop. Similar results were found in a study that had the same design but that substituted Stroop for a Flanker task, another perceptual conflict task said to recruit cognitive control (Hsu et al. 2021), demonstrating the generalisability of cross representational conflict adaptation.

What then distinguishes the non-replications of Kan et al. (2013) and the Hsu and Novick (2016) study that was replicated across two types of cognitive control measures (Flanker and Stroop)? There seem to be two key design distinctions. Hsu and colleagues presented the sentence comprehension task after the Stroop task, allowing them to assess comprehension and ensure resolution, and they used the visual world paradigm.

This last point gives rise to some concerns. The visual world paradigm is commonly used to study language processing; however, unlike eye-tracking-while-reading or self-paced reading paradigms, it also presents highly constraining non-linguistic information that may bias how we process the linguistic information. Therefore, in addition to sentence processing, it also reflects how participants make use of visual information to process

sentences (Huettig et al. 2011). For example, Tanenhaus et al. (1995) found that listeners can use this visual information to disambiguate ambiguous constructions faster than if they were not presented with a visual 'aid'. Likewise, any syntactic processing needs to be mapped back onto the visual display. Thus, there is a question as to whether the conflict in the Stroop task is directly transferring to syntactic conflict resolution or is having an effect during visual object processing downstream. If the transfer is to visual conflict rather than to syntactic conflict, which may be resolved via the parser alone, it would be consistent with other work that has argued conflict adaptation is representation-specific (Akçay and Hazeltine 2011; Egner et al. 2007; Forster and Cho 2014).

*1.5. The Current Study*

We have presented support for improved processing of garden-path sentences with experience either over a long (i.e., multiple weeks to a typical 1 h experiment) or short (i.e., trial) time scale. Different researchers tend to interpret the facilitation in garden path processing either solely within the parser or via support from a domain-general cognitive control mechanism. One limitation of the work that most clearly demonstrates a causal role of domain-general cognitive control facilitating syntactic disambiguation is that it is largely dependent on the visual world paradigm. Given that, the current study uses the same Stroop to sentence comprehension task order as Hsu and Novick (2016) but substitutes the visual world paradigm for a reading paradigm that excludes any visual information (and thus any non-linguistic visual conflict) in four experiments that had higher power (Exp. 1, *n* = 96; Exp. 2, *n* = 168; Exp. 3, *n* = 176; Exp. 4, *n* = 122) than the original studies on conflict adaptation that had less than half this number of participants (*n* = 41 in Kan et al. (2013); *n* = 23 in Hsu and Novick 2016; *n* = 26 in Hsu et al. 2021). We chose to use the RC-MC ambiguity in Experiments 1–3 as these sentences have already been shown to be sensitive to adaptation over an experiment (Fine et al. 2013; Fine and Jaeger 2016; Prasad and Linzen 2021) and critical support for the claim that domain-general conflict-monitoring mechanisms underlies both conflict adaptation and syntactic adaptation would be provided by their joint observation in the same data set with the same materials, task, and participants. Experiments 1 and 2 used self-paced reading and Experiment 3 used full-sentence reading to further improve the naturalness of the task and minimise the temporal interval between Stroop conflict resolution and syntactic disambiguation. The goal of the latter point is to help maximise the likelihood that domain-general cognitive control mechanisms would remain active. Although we replicated the ambiguity effect and syntactic adaptation in reading times in Experiments 1–3 and observed the Stroop effect in each experiment, we failed to find any evidence of conflict adaptation in any of the experiments. To ensure the lack of conflict adaptation was not due to the nature or difficulty of the reanalysis, Experiment 4 used the same goal-modifier ambiguity and filler sentence design (to avoid contiguous adaptation) used by Hsu and Novick (2016), again in self-paced reading to avoid visual object conflict. In Experiment 4, we find an ambiguity effect at the disambiguating region but no syntactic adaptation. Again, we find a Stroop effect but fail to find any conflict adaptation. We argue that most likely the conflict adaptation effect found in previous work is inherent to the visual world paradigm and is due to Stroop impacting visual object processing that interacts with language processing, rather than directly impacting language processing itself. The syntactic adaptation data are commensurate with both structural expectation accounts as well as reanalysis improvement most parsimoniously occurring within the parser. In the least, the results call for some domain-specific constraints on the underlying mechanism.

## 2. Experiment 1

In merging work on trial and contiguous adaptation to garden path processing, Experiment 1 modified the design of Fine and Jaeger (2016). Fine and Jaeger used the RC-MC ambiguity in self-paced reading and observed a decreasing ambiguity effect over exposure to RCs at the disambiguating region. We modified the design to cross a Stroop manipulation

(congruent vs. incongruent) with the ambiguity manipulation (ambiguous, unambiguous). Each trial included two tasks in sequence: Stroop and self-paced sentence reading. Based on prior work that argues for conflict monitoring assisting with ambiguity resolution, we expected a smaller ambiguity effect at the disambiguating region when the sentence is read directly after completing an incongruent Stroop stimulus compared to a congruent Stroop stimulus. No such difference is expected for the unambiguous sentence. We also expected to replicate a decreasing ambiguity effect at the disambiguating region with incremental exposure to RCs over the experiment. Further, if syntactic adaptation is due to learning via cognitive control mechanisms, then we might also find that as learning occurs over the experiment (syntactic adaptation), the conflict adaptation effect also dissipates as the conflict becomes less pronounced and thus cognitive control needs to be engaged to a lesser degree (i.e., a 3-way interaction between ambiguity, Stroop, and experimental item trial).

*2.1. Methods*

2.1.1. Participants

A total of 108 native speakers of English (56 women, 52 men) were recruited via the online recruitment platform Prolific (https://www.prolific.co) (accessed on 19 May 2020). Participants were screened within Prolific to exclude those with known hearing-, language-, or reading-related difficulties, and include those that were native English speakers, between the ages of 18 and 50 (Mage = 30.9, SDage = 8.1), and had normal or corrected-to-normal vision. We identified five participants who answered affirmatively to being colour-blind in our own internal check of these criteria via a short questionnaire at the start of the experiment; when contacted, these participants confirmed that this was selected by mistake. The study received ethical approval by UCL and all of the participants gave their informed consent before starting the study.

2.1.2. Materials

Gorilla Experiment Builder (https://www.gorilla.sc) (accessed on 28 March 2020) was used to create and host all 4 experiments. Materials, data and analysis code for this study are available at the APA's repository on Open Science Framework (OSF) and can be accessed at osf.io/mzd3w/. A within-participant design crossed 2 sentence ambiguity (ambiguous, unambiguous) and 2 Stroop congruency (incongruent, congruent). Each trial was made of a Stroop task followed by a sentence-reading task. This is a methodological improvement on previous conflict adaptation studies that simply interleaved the 2 sentence types with Stroop trials. The 36 critical sentences were adapted from Fine and Jaeger (2016) and manipulated the presence of a verb ambiguity (MC vs. RC). For example, *fed* in 6a. can be interpreted as either the main verb in the past tense or a past participle within a reduced relative clause. The parser's initial preference for it as the more probable main verb has to be re-analysed at the disambiguating *got a stomach*. The ambiguity is avoided in the unambiguous condition 6b. where the relative clause is explicitly introduced with *who were*:

6a. The sunburned boys fed the hot dogs got a stomach ache.

6b. The sunburned boys who were fed the hot dogs got a stomach ache.

Each sentence was followed by a yes/no comprehension question which targeted the correct, or final, interpretation of the sentence to verify it was eventually parsed correctly. For this, we created a set of questions that always required a 'yes' response. To avoid any contingencies between critical items and question type, and more importantly to ensure that the participants were not using a simple linear order strategy when answering them (e.g., by always paying attention only to the second half of the sentence), the comprehension questions were formulated in either the active or passive voice, and targeted either the main or the relative clause (N.B. question types were not manipulated, but assigned randomly to items):

7a. The sunburned boys fed the hot dogs got a stomach ache.

7b. Did the boys have a stomach ache?—MC active

8a.   The young technician taught the computer programme caught on right away.

8b.   Did someone teach the technician the programme?—RC active

9a.   Several angry workers wanted about low wages decided to file complaints.

9b.   Were complaints going to be filed by several angry workers?—MC passive

10a.  The experienced chef assured about the stove burnt his hand badly.

10b.  Was the chef assured about the stove?—RC passive

Critical trials were made up of a critical sentence preceded by a congruent or incongruent Stroop condition where participants had to indicate the colour of the word they saw appear on the screen. The design of the Stroop stimuli follows that used by Hsu and Novick (2016). Words were written in one of three colours—blue, green, and yellow. The font matched with the identity of the word on congruent trials, creating a total of 3 congruent item possibilities. Incongruent trials used a different set of colour words—brown, orange, and red—which were mismatched with one of the three colour responses, creating 9 incongruent item possibilities. The incongruent condition only gives rise to a representational conflict between the stimulus colour and word meaning and avoids response competition as there is no overlap between the identity of the word and any of the three response possibilities.

In addition to the critical items, 72 filler sentences were also adapted from Fine and Jaeger (2016). These sentences avoided any verbs with past tense/past participle-related ambiguity, nor did they include any relative clauses. Each filler item was also followed by a yes/no comprehension question and was also paired with a congruent or incongruent Stroop trial. However, as all 36 critical item questions required a 'yes' response, we created 54 filler questions to require a 'no' response to balance the proportion of yes/no responses across all items. Similarly, only 18 filler sentences were preceded with a Stroop trial while the other 54 had a sentence–Stroop pairing, resulting in an equal number of each pairing type across all items, which reduced the predictability of the upcoming task. Across the experiment, there was an equal number of congruent and incongruent Stroop trials, with the 3 congruent and 9 incongruent Stroop item possibilities balanced for each Stroop trial type.

We used a Latin square design to create 4 experimental lists where every ambiguous and unambiguous version of the sentence appeared with a congruent and incongruent Stroop condition across four lists. This helped avoid any pairing confounds—an improvement on Hsu and Novick's (2016) design which only included two lists that counterbalanced ambiguity but not its pairing with Stroop condition. Further, we created 2 unique pseudo-random orders that were then also reversed to control for any correlation between item identity and presentation order. This resulted in a total of 16 lists when applied to each of the 4 experimental lists. The pseudo-random order was inspired by the one used by Fine and Jaeger (2016) and followed the following rules: no more than three subsequent items with the same comprehension question response (yes/no); no more than three of the same Stroop trials in a row (congruent/incongruent); no more than three subsequent items with the same ambiguity; and no more than two experimental items in a row.

### 2.1.3. Procedure

Participants provided informed consent and then completed a short demographic questionnaire. Following Hsu and Novick's (2016) design, the participants then practised matching the Stroop font colours (as colour blocks) to their correct response buttons. The participants were asked to use the arrow buttons located in the lower right-hand corner of the keyboard and use their right index finger on the 'left arrow' to respond BLUE, middle finger on the 'up arrow' to respond GREEN, and ring finger on the 'right arrow' to respond YELLOW. After completing 18 colour patch trials, where they had to identify the colours of the squares presented in the centre of the screen, they completed 72 standard Stroop trials. A central fixation cross was displayed for 500 ms at the start of each practice trial, with 100 ms of blank screen appearing before and after it. After participants made their responses, feedback was presented for 300 ms. Feedback was only provided in the practice

trials but not during the actual experimental trials. The patch or colour word disappeared after 1500 ms if no response was made.

Following this, they completed 8 practice trials of a sequential Stroop and sentence comprehension task before starting the experiment. Sentences were read in a self-paced moving window display (Just et al. 1982). Self-paced reading is a resource-light method that has been shown to be a reliable measure of sentence processing even when administered online (Enochson and Culbertson 2015; Keller et al. 2009) The sentences were blackened out at the start of the trial (with the exception of the first word of the sentence) and each press of the spacebar (with their left thumb or index finger) revealed the following word while (re)masking the preceding word. Pressing the space bar after the last word would reveal the associated comprehension question. Comprehension questions were presented in full (not self-paced). Participants had to use their left hand to press the 'left arrow' to respond 'yes' and the 'right arrow' key to respond 'no'. Fingers from both of their hands had to be kept on the specified keys for the duration of the experiment.

A central fixation cross appeared for 500 ms, with 100 ms of blank screen displayed immediately before and after, preceding either a Stroop or Sentence comprehension task. Stroop trials timed out after 1500 ms without a response triggering the next task; words in the sentence-reading task did not have a time-out limit. The 108 critical and filler items were presented in a pseudo-randomised order in three blocks of 36 items each, with optional breaks after blocks 1 and 2. Although the breaks were not timed, the entire experiment would time out after 115 min. Figure 1 illustrates these trial dynamics.

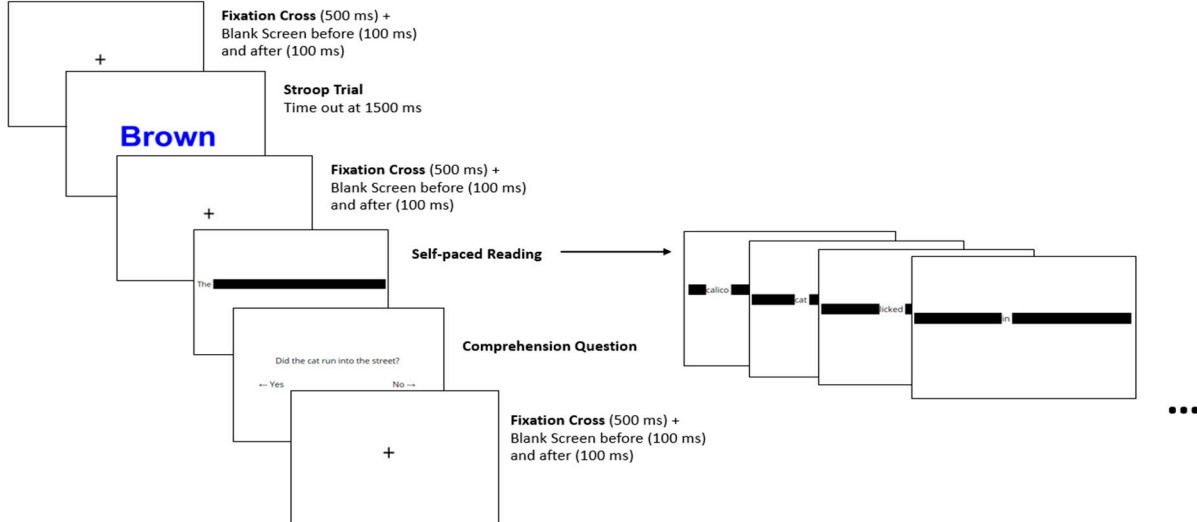

**Figure 1.** Trial Dynamics for Experiments 1, 2, 3, and 4. Each item consisted of a Stroop and a sentence reading pair, with each sentence followed by a comprehension question. This order was also reversed in some filler items. Trial dynamics were the same in each experiment except for Experiment 3 where self-paced reading was replaced with full-sentence presentation.

2.1.4. Exclusions

Participants scoring below 80% on either Stroop or filler comprehension questions were excluded, resulting in the exclusion of 12 participants. The average accuracy of the 96 remaining participants was 93% for both Stroop and filler comprehension questions, indicating good levels of attention and performance.

Practice sentences were excluded from the data analysis. The first word of every sentence was also excluded as they appeared unmasked and their processing time could be confounded by the recovery from the preceding task. Any trials (Stroop + sentence) with sentences with RTs over 20,000 ms were also excluded as it likely reflects a lapse in attention and may bias the effect of the preceding Stroop trial on sentential conflict. This resulted in the exclusion of 0.51% or 53 of all trials (critical and filler). Any words with RTs

less than 100 ms or more than 2000 ms were also excluded, as is standard in relevant prior work. This resulted in the exclusion of 0.25% of all words, comparable to the <1% excluded in Fine et al. (2013) and 0.4% in Fine and Jaeger (2016). We did not exclude any trials with incorrect comprehension questions as they do not necessarily indicate that the sentences were not parsed (Novick et al. 2014; but see Harrington Stack et al. 2018).

2.1.5. Data Analysis

Mixed-effect models were fitted using the lme4 package (version 1.1-30) with the LmerTest extension package (version 3.1-3) used to view the degrees of freedom and *p*-values in R—4.2.2. Any interactions were decomposed using the emmeans package (version 1.8.0). For the best generalisability of the model, a maximal random effect structure justified by the design is typically recommended (Barr et al. 2013). However, as our models are relatively complex, none of them converged with a maximal random effect structure. Following the processing described in Dempsey et al. (2020), the variables that contributed the least variance were removed one by one until the model converged. For the full structures of all the models, please refer to the R script in the Supplementary Materials.

*Stroop Reaction Time.* Any timed-out Stroop trials (2.5% of all experimental Stroop trials) were excluded from this analysis. A linear mixed-effects model for Stroop RTs included the fixed effect of Stroop and log-transformed item order (filler and experimental items: 1–108; referred to as log (Trial) from here on). Item order effect was included to control for any task learning effects that may bias the experimental effects. This effect was log-transformed as Fine et al. (2010) found models with log-transformed item presentation order to be better predictors of task adaptation than its linear effect, arguing that task adaptation is a log-linear rather than simply a linear effect. A Stroop item random effect was included to account for the relative difference in performance between the 12 Stroop word-colour pairings (3 congruent and 9 incongruent) as well as a participant random effect. Deviation coding was used to set Stroop contrasts to −1 for incongruent and 1 for congruent Stroop.

*Reading Time Data.* Critical sentences were divided into the same five regions as in previous studies using these sentences (Fine et al. 2013; Fine and Jaeger 2016; Yan and Jaeger 2020), which included *subject*, (*relativiser*), *ambiguous region*, *disambiguating region*, *final word* (spill): Sunburned children/(who were)/fed the hot dogs/got a stomach/ache.

As in previous syntactic adaptation studies (Fine et al. 2013; Fine and Jaeger 2016), raw RTs were residualised to avoid word-length effects. Raw RTs (for both filler and experimental items) were first log transformed to normalise data distribution and then regressed onto the character length of each word within a model which also included a participant random effect. The residuals of this model, which excluded word length effects, were then averaged for each sentence region. This was used as the predicted measure in all the subsequent analyses.

A separate linear mixed-effect model was fitted for the subordinate verb, ambiguous NP, disambiguating and spill regions. Although only the disambiguating and ambiguous regions will be discussed here, results for each region are summarised in Table 1. Each of these models included fixed effects of sentence ambiguity (ambiguous vs. unambiguous), Stroop (congruent vs. incongruent), experimental item trial (as a continuous predictor coded 1–40) and their interactions. Similar to the Stroop model, a fixed effect for log-transformed items was also included to control for task adaptation effects. Log (Trial) and experimental item trial were also centred around their means to reduce collinearity with higher-order interactions. Participants and items were also included as random effects. Its maximal structure which included ambiguity and Stroop random slopes was minimised until convergence by using the technique described by Dempsey et al. (2020).

**Table 1.** Coefficients and t Values for Residual log RTs for each Predictor (Rows) at Each Sentence Region (columns) in Experiment 1.

| Predictor | Ambiguous Region | | Disambiguating Region | | Spill (Final Word) | |
|---|---|---|---|---|---|---|
| | β | t | β | t | β | t |
| Ambiguity (=amb) | **0.01** | **2.56 *** | **0.02** | **3.94 ***** | **0.02** | **3.30 **** |
| Stroop(=cong) | <0.01 | 0.1 | <0.01 | 0.42 | <−0.01 | −0.70 |
| Exp. Item trial | **<−0.01** | **−2.72 **** | **<−0.01** | **−1.97 *** | <0.01 | 0.33 |
| Ambiguity:Stroop | **<−0.01** | **−2.23 *** | <−0.01 | −0.53 | <0.01 | 0.12 |
| Ambiguity:Exp. Item trial | <0.01 | 0.66 | <−0.01 | −1.71 | **<−0.01** | **−3.09 **** |
| Stroop:Exp. Item trial | <−0.01 | −0.59 | <0.01 | 0.41 | <0.01 | 0.18 |
| Amb:Stroop:Exp. Item tr. | <0.01 | 0.02 | <0.01 | 0.23 | <−0.01 | −0.92 |
| Log(trial) | **−0.06** | **−5.29 ***** | **−0.07** | **−4.95 ***** | **−0.12** | **−4.93 ***** |

Note. * $p < 0.05$; ** $p < 0.01$; *** $p < 0.001$. Bolded values indicate significant effects.

Deviation coding was used to set Stroop contrasts to −1 for incongruent and 1 for congruent, and ambiguity contrasts at 1 for ambiguous and −1 for unambiguous, thus ensuring that the mean of each of these fixed effects equalled zero, allowing us to interpret any significant effect as the main effect of that variable.

*Comprehension Accuracy.* A generalised linear mixed-effect model with a binomial distribution for comprehension accuracy included all the same fixed and random effects as the model for sentence RTs. However, an effect of log(Trial) was not included since responding to questions is not a novel task that we can adapt to in the same way that we can adapt to the 'unnatural' Stroop or self-paced reading tasks. In other words, the reading/reaction times reflect both the cognitive processing as well as the 'mechanical reaction' to the task (i.e., button press) that will improve with practice but does not reflect any improvement in processing; however, the accuracy measure does not reflect any extraneous reaction to the task itself (there is no time limit to answer the questions), and any increase in accuracy over the course of the experiment would indicate an improvement in the processing mechanisms. We also used a bound optimization by quadratic approximation optimiser (BOBYQA) set to 100 thousand iterations available in the lme4 package.

## 2.2. Results

### 2.2.1. Stroop Reaction Time

There was a standard effect of Stroop (β = −34.71, SE = 12.78, df = 9.95, t = −2.72, $p = 0.022$), where RTs for congruent Stroop trials were significantly shorter than RTs for incongruent trials. In addition, there was a main effect of log(trial) (β = −19.45, SE = 2.03, df = 9895.66, t = −9.59, $p < 0.001$), where RTs got significantly shorter as the experiment went on.

### 2.2.2. Sentence RT

The results for each of the regions are summarised in Table 1. However, the discussion will focus on the two regions of interest: ambiguous and disambiguating regions.

*Ambiguous Region.* There was a standard effect of ambiguity (β = 0.01, SE < 0.01, df = 34.08, t = 2.56, $p = 0.015$). There were also main effects of experimental item trial (β = −0.003, SE < 0.01, df = 210.14, t = −2.72, $p = 0.007$) and log(trial) (β = −0.06, SE = 0.01, df = 176.80, t = −5.29, $p < 0.001$), where RTs for both sentence types progressively decreased over the course of the experiment.

There was also an interaction between Stroop and ambiguity (β = −0.006, SE < 0.01, df = 3265.67, t = −2.23, $p = 0.026$). However, the interaction was not at the predicted region nor did it follow the pattern found by Hsu and Novick (2016) at their disambiguating region. Although they found that only ambiguous sentences were affected by the type of the preceding Stroop trial and not unambiguous sentences, we found that neither the ambiguous ($p = 0.131$) nor unambiguous sentences ($p = 0.100$) were differentially affected by the type of Stroop condition that preceded it. However, we found that although there was an ambiguity effect in post-incongruent trials (β = 0.04, SE = 0.01, df = 70.2, t = 3.37, $p = 0.001$), the ambiguity effect disappeared in post-congruent trials ($p = 0.365$) (Figure 2).

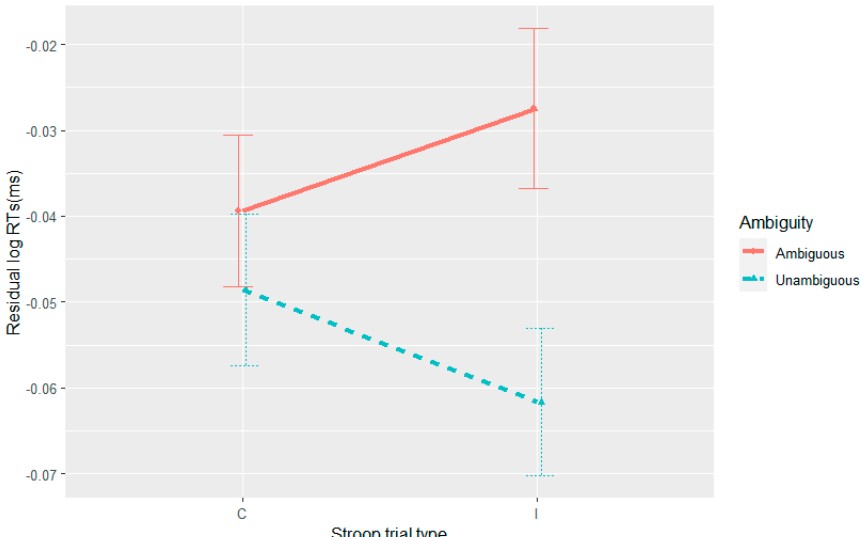

**Figure 2.** Ambiguity by Stroop Interaction at the Ambiguous Region in Experiment 1. Note that the plot shows an ambiguity effect for items preceded by incongruent Stroop (right side of the graph). This same ambiguity effect is absent on trials preceded by congruent Stroop trials (where the error bars for ambiguous and unambiguous items overlap—left side of the graph). The figure illustrates mean residual log reading times with the error bars representing 95% confidence intervals.

*Disambiguating Region.* There was a standard effect of ambiguity (β = 0.02, SE < 0.01, df = 45.22, t = 3.94, *p* < 0.001). There were also main effects of experimental item trial (β = −0.002, SE < 0.01, df = 546.57, t = −1.97, *p* = 0.049) and log(trial) (β = −0.07, SE = 0.01, df = 484.66, t = −4.95, *p* < 0.001), following the same pattern as in the previous region. However, we found no predicted interaction between Stroop and ambiguity (*p* = 0.684) (Figure 3).

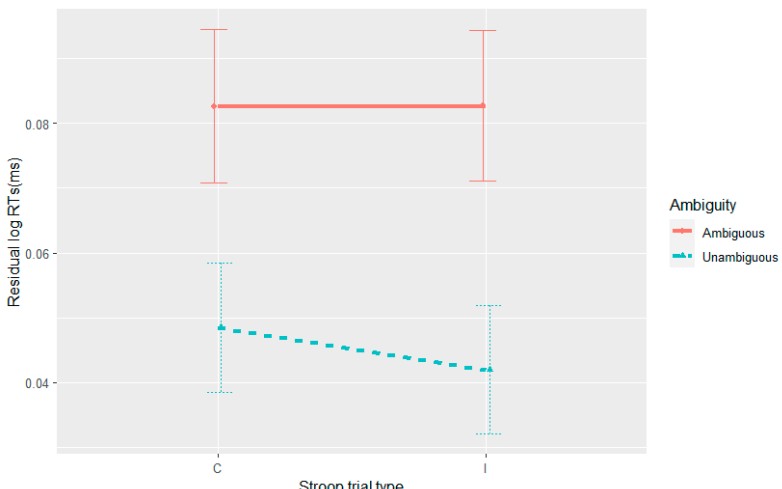

**Figure 3.** Ambiguity by Stroop Interaction at the Disambiguating Region in Experiment 1. Note that the plot shows a lack of interaction between Stroop and ambiguity effects. The figure illustrates mean residual log reading times with the error bars representing 95% confidence intervals.

Although the interaction between ambiguity and experimental item trial was not significant (β < −0.001, SE < 0.01, df = 3139.04, t = −1.71, *p* = 0.087), the numerical trend replicated the one found in previous studies on syntactic adaptation (Fine et al. 2013; Fine and Jaeger 2016) as can be seen in Figure 4. The absence of significance is likely due to a lack of statistical power (Harrington Stack et al. 2018) as we find the same interaction at significance at the final word region (β = −0.002, SE < 0.01, df = 3168.08, t = −3.09, *p* = 0.002),

where the ambiguity effect gets significantly smaller as the experiment progresses, an effect also found at the same region in previous studies (Fine et al. 2013; Fine and Jaeger 2016).

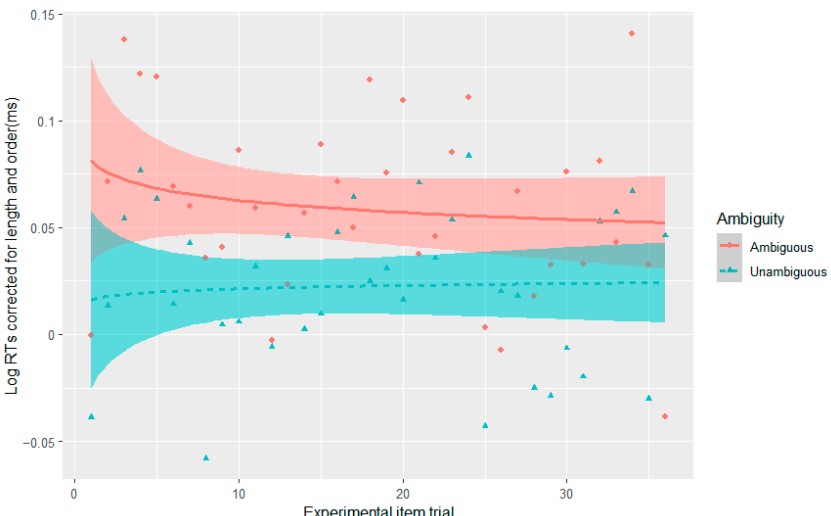

**Figure 4.** Change in Ambiguity Effect at Disambiguating Region as a Function of Number of Experimental Items Seen for Experiment 1. Note that the plot demonstrates a numerical decrease in ambiguity effect (the difference in RRTs between ambiguous (red line) and unambiguous (green line) sentences). The trendline is presented log-linearly. Log RTs were residualised for word length and trial order to account for task adaptation to as best as possible approximate the model that included trial order as a fixed factor.

### 2.2.3. Comprehension Question Accuracy

A summary of accuracy results for all 4 experiments can be found in Table 2.

We found a standard ambiguity effect for question accuracy ($\beta = -0.30$, SE = 0.11, $z = -2.61$, $p = 0.009$).

**Table 2.** Coefficients and z-values for Accuracy (0–1) for Each Predictor (Rows) for Experiments 1–4.

| Predictor | Experiment 1 | | Experiment 2 | | Experiment 3 | | Experiment 4 | |
|---|---|---|---|---|---|---|---|---|
| | β | z | β | z | β | z | β | z |
| Ambiguity (=amb) | **−0.30** | **−2.61 \*\*** | **−0.48** | **−4.46 \*\*\*** | **−0.30** | **−4.75 \*\*\*** | **−0.23** | **−2.68 \*\*** |
| Stroop(=cong) | 0.01 | 0.21 | I−C: −0.13 **N−I: 0.31** | −1.06 **2.42 \*** | <−0.01 | −0.16 | 0.04 | 0.82 |
| Exp. Item trial | <0.01 | 0.41 | <0.01 | 0.92 | **<0.01** | **2.69 \*\*** | 0.01 | 1.61 |
| Question Type | NA | NA | NA | NA | **0.28** | **2.09 \*\*** | NA | NA |
| Ambiguity:Stroop | −0.12 | −1.75 | I−C: 0.08 N−I: −0.12 | 0.68 −0.92 | **0.12** | **2.77 \*\*** | −0.04 | −0.75 |
| Ambiguity:QT | NA | NA | NA | NA | 0.03 | 0.54 | NA | NA |
| Stroop:QT | NA | NA | NA | NA | 0.04 | 1.05 | NA | NA |
| Ambiguity:Exp. Item trial | <0.01 | 0.60 | <0.01 | 0.97 | <−0.01 | −0.71 | **0.01** | **3.44 \*\*\*** |
| Stroop:Exp.Item trial | 0.01 | 1.95 | I−C: <0.01 N−I: −0.01 | 0.09 −0.85 | <−0.01 | −0.91 | **<0.01** | **2.51 \*** |
| QT:Exp.Item trial | NA | NA | NA | NA | <−0.01 | −1.60 | NA | NA |
| Amb:Strp:QT | NA | NA | NA | NA | <−0.01 | −0.80 | NA | NA |
| Amb:Strp:Exp. Item tr. | **−0.02** | **−2.59 \*\*** | I−C: −0.01 N−I: 0.1 | −0.77 0.92 | <0.01 | 0.67 | **<−0.01** | **−1.97 \*** |
| Amb:QT:Exp. Item tr. | NA | NA | NA | NA | <−0.01 | −0.99 | NA | NA |
| Strp:QT:Exp. Item tr. | NA | NA | NA | NA | **<0.01** | **2.41 \*** | NA | NA |
| Amb:Strp:QT:Exp. It. Tr. | NA | NA | NA | NA | <−0.01 | −1.62 | NA | NA |

Note. * $p < 0.05$; ** $p < 0.01$; *** $p < 0.001$. Bolded values indicate significant effects.

We also found an unexpected three-way interaction between ambiguity, Stroop, and experimental item trial ($\beta = -0.02$, SE < 0.01, z = $-2.58$, $p = 0.010$). After decomposing the interaction, we found that the interaction was not from the expected change in accuracy to the ambiguous sentence probe, as it stays constant on post-congruent ($p = 0.833$) and post-incongruent ($p = 0.368$) trials, but accuracy decreases for post-incongruent unambiguous probes ($\beta = -0.03$, SE = 0.01, z = -2.26, $p = 0.024$), with accuracy to the unambiguous sentences on post-congruent Stroop remaining stable ($p = 0.059$). This pattern does not follow any predicted by the conflict monitoring theory (or any other cognitive control/Stroop theories) and will therefore not be discussed any further.

*2.3. Discussion*

While we replicated the syntactic adaptation effect at the final word and observed it numerically at the disambiguating region, we did not find that the preceding Stroop impacted the ambiguity effect that was observed at the disambiguating region. This was not due to a failure of Stroop to engage cognitive control mechanisms, as we did see the classical Stroop effect. As we did not observe conflict adaptation even numerically, it does not seem to be an issue of power.

Furthermore, we replicated the ambiguity effect at the ambiguous region that was observed by Fine and Jaeger (2016) and MacDonald et al. (1992). Unexpectedly, we found that congruent Stroop eliminated this effect at the ambiguous region; however, it is unclear whether incongruent Stroop had any impact on the size of the ambiguity effect. This finding is not consistent with the conflict monitoring theory, whereby processing conflicting representations is supposed to facilitate the processing of subsequent ones. The theory neither predicts Stroop having an effect at the ambiguous region (that lacks any processing conflict) nor congruent Stroop having an effect at any region. However, there is work outside of sentence processing that has argued that adaptation across Stroop trials is not only due to a preceding incongruent Stroop but also due to the effect of a preceding congruent Stroop (Compton et al. 2012; Gratton et al. 1992; Lamers and Roelofs 2011). In order to better assess whether this is the case, Experiment 2 will attempt to replicate the effect and further understand how incongruent and congruent Stroop contribute to the effect by including a neutral Stroop condition for comparison.

## 3. Experiment 2

Given the novel finding at the ambiguous region in Experiment 1, whereby congruent Stroop eliminated the ambiguity effect, but was still observed in post-incongruent Stroop trials, we attempted to replicate it in Experiment 2. To better understand the specific contributions of incongruent and congruent trials in this region we included a third neutral Stroop condition for comparison. Given syntactic adaptation was not significant at the disambiguating region, we also increased the statistical power.

*3.1. Methods*

3.1.1. Participants

A total of 188 native speakers of English (105 women, 83 men, Mage = 32.0, Sdage = 8.5) were recruited via the online recruitment platform Prolific (https://www.prolific.co) (accessed on 22 January 2021). In addition to the previous exclusion criteria, anyone who participated in Experiment 1 was excluded from participation in Experiment 2. Our internal check identified four participants who answered affirmatively to being colour-blind and three participants who stated that they were not native English speakers; all of these confirmed that this was selected by mistake when contacted.

3.1.2. Materials

The materials used were identical to the ones used in the previous study, with the exception of an additional neutral Stroop condition, expanding the design to a 2 sentence ambiguity (ambiguous, unambiguous) by 3 Stroop (congruent, incongruent, neutral).

The most common implementation of a neutral Stroop is to replace the standard colour words with another word unrelated to colour, for example an animal name (e.g., dog, cat, and mouse in Compton et al. 2012). However, since the neutral trial in our experiment needed to act as a baseline for testing the impact of congruent vs. incongruent trials on the resolution of a sentential conflict, it needed to be neutral in two senses: to avoid any overlap between the two features of the Stroop stimulus (i.e., the font colour and the string) and to avoid activating any language related mechanisms that could bias our understanding of its effect on subsequent language processing. Therefore, instead of using non-colour words, which would still automatically engage the reading process and activate conflicting semantics (Kinoshita et al. 2017) and could generate some lexical conflict, we decided to use non-linguistic symbols to create neutral stimuli. We used five recurring hash/pound symbols (#####) in one of the three colour response options—blue, green, and yellow. We hypothesised that after being exposed to several such trials the participants would quickly learn to ignore the string itself (as it was always the same, unlike the colour words), and automatically focus only on the colour in which it is presented. This would help avoid any processing associated with reading, as well as help avoid any congruency or incongruency effects (as there would be no real association between the two features of the stimulus), allowing us to understand any effects that stem from processes associated specifically with congruent or incongruent trials.

To keep the number of each Stroop trial type balanced, the number of practice trials had to be increased to 81 as 72 is not divisible by 27. However, the number of experimental Stroop trials (and critical and filler sentences) remained the same—108, as this still allowed an equal number of each Stroop type. Similarly, the number of unique Stroop–sentence pairs increased to 9.

Finally, we substituted the pseudo-randomised order of presentation with a fully randomised one. Full randomisation of items provides a better way to control for any presentation order effects (a unique presentation order for each participant vs. just 4 orders in the previous experiment). Any contingency effects due to a randomised order that we aimed to avoid with a pseudo-random order (e.g., more than 2 experimental trials in a row) should be smoothed over with a large enough sample size.

### 3.1.3. Procedure

The procedure was exactly the same as in the previous experiment.

### 3.1.4. Exclusions

A total of 20 participants were excluded as they scored below 80% on filler questions and/or Stroop trials. Although there were more participants excluded in the second experiment, proportionally it is the same in both experiments (11% of total sample size). The remaining participants were on average 93% accurate on both Stroop trials and filler questions.

We excluded 0.31% or 57 trials with sentences that took longer than 20,000 ms to read and a further 0.31% of all words that took less than 100 ms or more than 2000 ms to read.

### 3.1.5. Data Analysis

The same analyses were performed as for Experiment 1. However, deviation coding was replaced with successive difference contrast coding, where each level was compared to a previous one in succession to accommodate the third Stroop condition. By predefining the order of Stroop levels as congruent–incongruent–neutral, we obtain results for two comparisons: incongruent vs. congruent testing for the standard Stroop effect, and incongruent vs. neutral testing for the impact of incongruent trials relative to the 'baseline' neutral trials. This comparison was crucial to verify the conflict adaptation hypothesis, as interpreted by Hsu and Novick (2016), where any effect stems specifically from the incongruent trials.

### 3.2. Results

#### 3.2.1. Stroop Reaction Time

There was a main effect of Stroop, with a significant incongruent–congruent contrast ($\beta$ = 66.22, SE = 18.98, df = 11.62, t = 3.49, $p$ = 0.005) indicative of a standard Stroop effect, and a significant incongruent–neutral contrast ($\beta$ = −51.61, SE = 18.98, df = 11.62, t = −2.72, $p$ = 0.019), where neutral trials were completed faster than incongruent ones. This difference between these three levels can be seen in Figure 5.

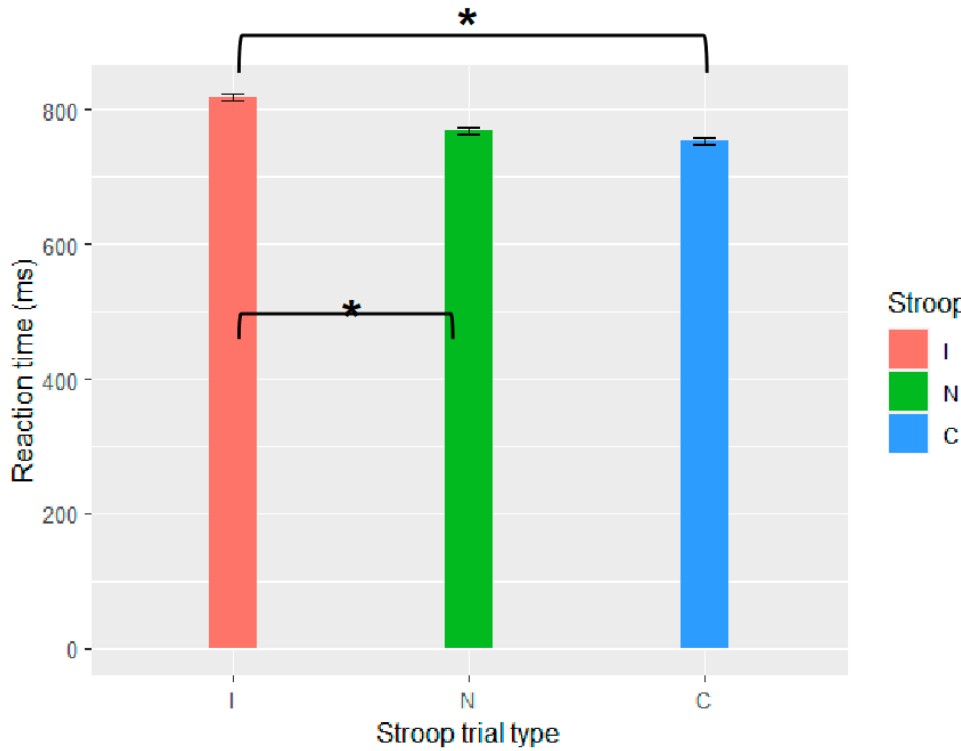

**Figure 5.** Difference in Reaction Time between Three Stroop Trial Types. Note that the graph shows a higher RT for incongruent trials relative to neutral and congruent trials. The error bars represent 95% confidence intervals. * indicates a significant difference between connected conditions.

#### 3.2.2. Sentence RT

A summary of results for each of the three regions of interest is presented in Table 3.

**Table 3.** Coefficients and t-values for Residual log RTs for Each Predictor (Rows) and Each Sentence Region (Columns) in Experiment 2.

| Predictor | Ambiguous Region | | Disambiguating Region | | Spill (Final Word) | |
|---|---|---|---|---|---|---|
| | β | t | β | t | β | t |
| Ambiguity (=amb) | **0.02** | **7.11 ***** | **0.02** | **8.14 ***** | **0.02** | **3.92 ***** |
| Stroop(I-C) | <−0.01 | −1.71 | <−0.01 | −0.93 | <−0.01 | −0.54 |
| Stroop (N-I) | <0.01 | 0.96 | <0.01 | 1.04 | <0.01 | 0.02 |
| Exp. Item trial | **<−0.01** | **−4.68 ***** | **<−0.01** | **−3.82 ***** | **<−0.01** | **−4.15 ***** |
| Ambiguity:Stroop(I-C) | <0.01 | 0.35 | <0.01 | 0.82 | 0.01 | 1.15 |
| Ambiguity:Stroop(N-I) | <0.01 | 0.76 | <0.01 | 1.00 | <0.01 | 0.10 |
| Ambiguity:Exp. Item trial | <0.01 | 0.57 | **<−0.01** | **−3.96 ***** | **<−0.01** | **−2.97 **** |
| Stroop(I-C):Exp. Item trial | <0.01 | 1.15 | <−0.01 | −0.04 | <0.01 | 1.94 |
| Stroop(N-I):Exp. Item trial | <−0.01 | −1.44 | <−0.01 | −1.44 | <0.01 | 0.28 |
| Amb:Str(I-C):Exp. Item tr. | <0.01 | 0.21 | <−0.01 | −1.81 | <−0.01 | −0.68 |
| Amb:Str(N-I):Exp. Item tr. | <0.01 | 0.78 | <0.01 | 1.70 | <0.01 | 0.08 |
| Log(trial) | **−0.05** | **−9.29 ***** | **−0.06** | **−11.75 ***** | **−0.06** | **−6.81 ***** |

Note. ** $p$ < 0.01; *** $p$ < 0.001. Bolded values indicate significant effects.

*Ambiguous Region.* There was a standard effect of ambiguity (β = 0.02, SE < 0.01, df = 33.09, t = 7.11, *p* < 0.001) as well as main effects of experimental item trial (β = −0.002, SE < 0.01, df = 5983.45, t = −4.68, *p* < 0.001) and log(trial) (β = −0.05, SE = 0.01, df = 5964.67, t = −9.29, *p* < 0.001) following the same pattern as in Experiment 1.

Contrary to Experiment 1, there was no interaction between ambiguity and Stroop for the incongruent–congruent (*p* = 0.724) contrast, nor was there one for the neutral–incongruent contrast (*p* = 0.450).

*Disambiguating Region.* There was a standard effect of ambiguity (β = 0.02, SE < 0.01, df = 165.32, t = 8.14, *p* < 0.001). There were also main effects of experimental item trial (β = −0.002, SE < 0.01, df = 5920.08, t = −3.82, *p* < 0.001) and log(trial) (β = −0.06, SE < 0.01, df = 5930.21, t = −11.75, *p* < 0.001), once again following the same pattern as before. Replicating the results of Experiment 1, we found no interaction between Stroop and ambiguity for the incongruent–congruent contrast (*p* = 0.410) nor neutral–incongruent one (*p* = 0.319) (Figure 6).

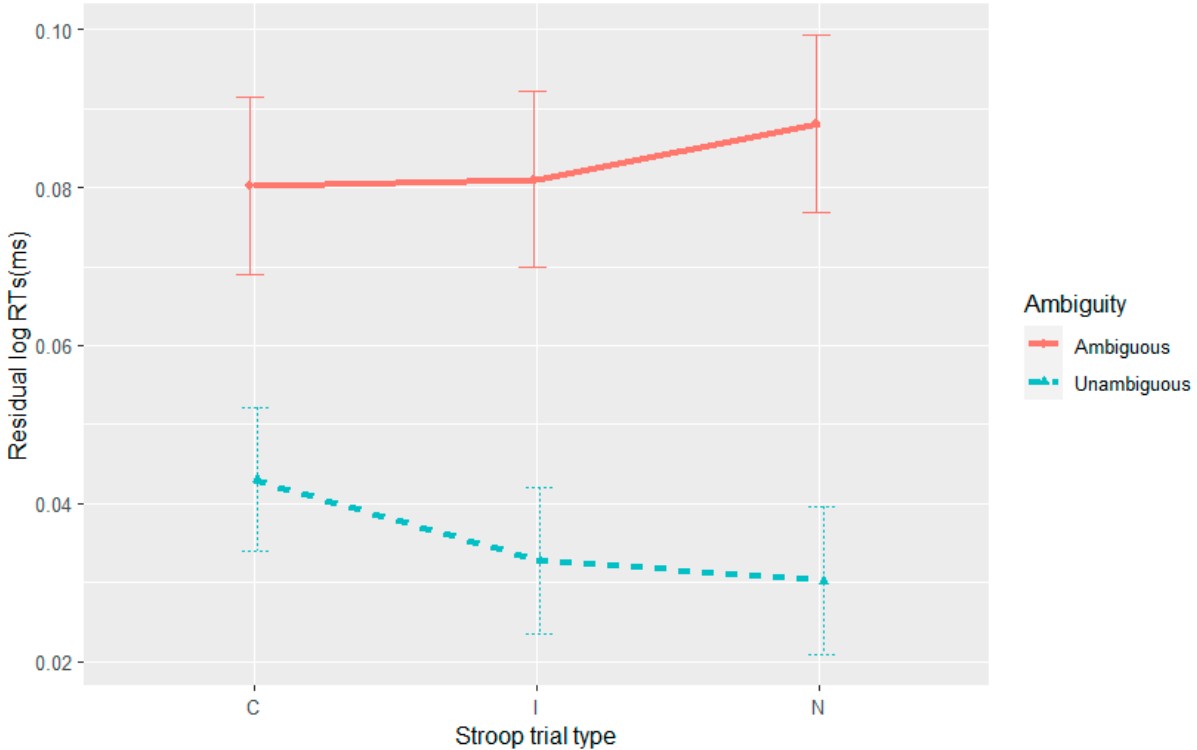

**Figure 6.** Ambiguity by Stroop Interaction at the Disambiguating Region in Experiment 2. Note that the plot shows a lack of interaction between Stroop and ambiguity effects. The figure illustrates mean residual log reading times with the error bars representing 95% confidence intervals.

There was also a significant interaction between ambiguity and experimental item trial (β = <−0.001, SE < 0.01, df = 5725.87, t = −3.96, *p* < 0.001), where the ambiguity effect got smaller as the experiment progressed as the result of a steady decrease in ambiguous RTs (β = −0.003, SE < 0.01, df = 5895, t = −5.17, *p* < 0.001), while unambiguous RTs remained stable (*p* = 0.078) (Figure 7).

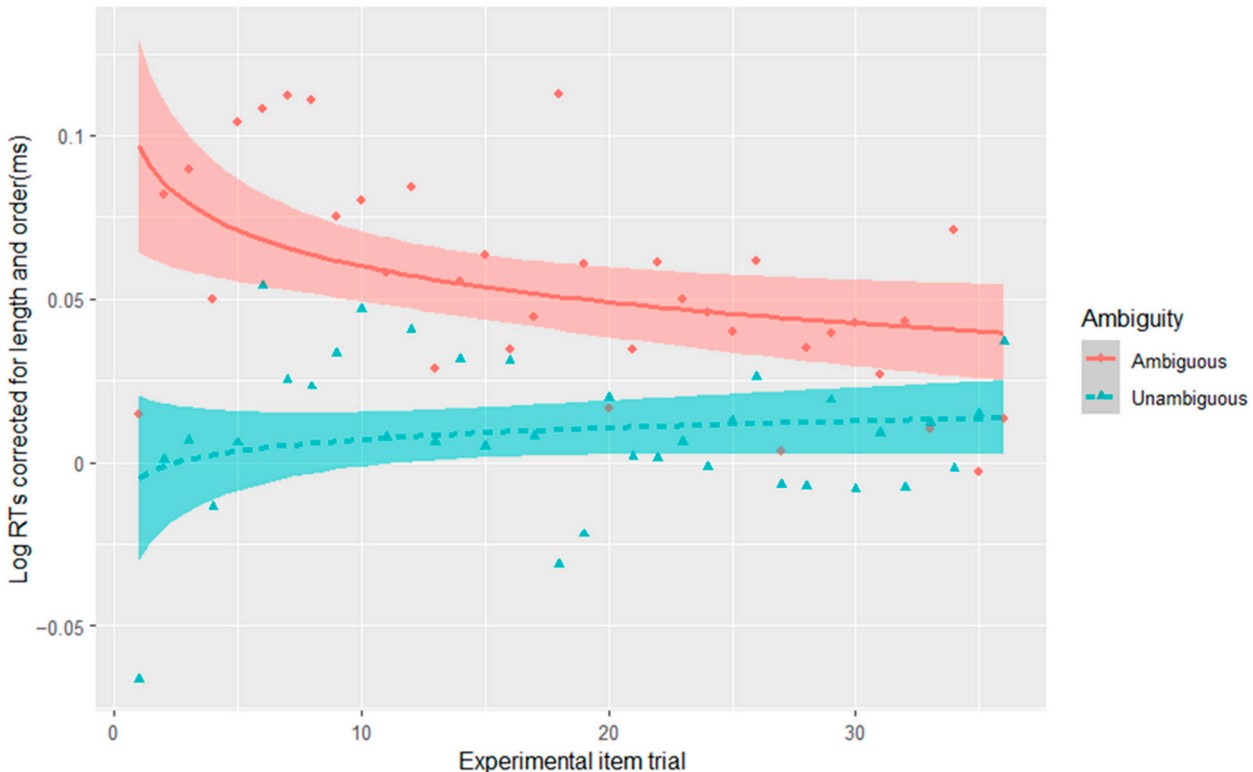

**Figure 7.** Change in Ambiguity Effect at Disambiguating region as a Function of Number of Experimental Items Seen for Experiment 2. Note that the plot demonstrates a decrease in ambiguity effect (the difference in RRTs between ambiguous (red line) and unambiguous (green line) sentences). The trendline is presented log-linearly. Log RTs were residualised for word length and trial order to account for task adaptation to as best as possible approximate the model that included trial order as a fixed factor.

### 3.2.3. Comprehension Question Accuracy

There was a standard effect of ambiguity ($\beta = -0.48$, SE = 0.11, z = $-4.46$, $p < 0.001$). There was also a main effect of Stroop ($\beta = 0.31$, SE = 0.13, z = $-2.42$, $p = 0.016$), where comprehension questions that were preceded by neutral Stroop were on average more accurate than those preceded by incongruent Stroop. We did not replicate the three-way interaction between ambiguity, Stroop, and experimental item trial found in Experiment 1 for either of the two contrasts (I-C: $p = 0.440$; N-I: $p = 0.357$).

### 3.3. Discussion

We did not replicate congruent Stroop's elimination of the ambiguity effect at the ambiguous region, thus its observation in Experiment 1 was most likely a spurious result. We also again failed to observe conflict adaptation at the disambiguating region, statistically or even numerically. With increased statistical power, however, we successfully replicated syntactic adaptation at the disambiguating region.

A concern for our inability to observe conflict adaptation is whether too much time passes from the Stroop task to the disambiguating region to allow for continual activation of conflict monitoring mechanisms. Hsu and Novick's (2016) study that successfully showed incongruent Stroop facilitating garden path processing used spoken sentences as stimuli, given the visual world paradigm. As we are using self-paced reading, a motor response is required for each word, and likely increases the duration from Stroop to disambiguation compared to Hsu and Novick (2016). Thus, it may be that in our Experiments 1 and 2, the mechanisms had dissipated further, or completely, to not have an observable effect. Indeed, in looking at the audio recordings of the sentence stimuli from the Supplementary Materials of Hsu and Novick (2016) we find they had a shorter duration to disambiguation

than in our Experiments 1 and 2 (see Table 4). To help address this, Experiment 3 will change from self-paced reading to full-sentence presentation while also benefitting from a more natural task.

**Table 4.** Comparison in ISI (Stroop—Disambiguation and Stroop—End of Sentence) between Experiments 1–4 and Hsu and Novick (2016).

|  | Experiment 1 | Experiment 2 | Experiment 3 | Experiment 4 | Hsu and Novick (2016) |
|---|---|---|---|---|---|
| Ambiguous sentence |  |  |  |  |  |
| Full sentence | 5492 ms | 5520 ms | 5007 ms | 4607 ms | 2957 ms |
| Until disambiguation | 3659 ms | 3685 ms | NA | 3107 ms | 2625 ms |
| Unambiguous sentence |  |  |  |  |  |
| Full sentence | 6029 ms | 5970 ms | 4805 ms | 4896 ms | 3198 ms |
| Until disambiguation | 4321 ms | 4275 ms | NA | 3479 ms | 2835 ms |

Note. Total time includes an additional 700 ms (fixation cross + blank screens) for Experiments 1–4 and an additional 800 ms (fixation cross + delay before start of audio) for Hsu and Novick (2016).

## 4. Experiment 3

Experiment 3 used the same design as Experiment 1 but with two modifications: substitution of self-paced reading with full-sentence reading and inclusion of a comprehension question manipulation. Full-sentence reading has two benefits towards our goals: potentially reducing the duration from Stroop conflict to sentence conflict and providing a more natural task. The second benefit is also one for syntactic adaptation, as it will eliminate the learning required for self-paced reading.

In our previous two experiments, we included a mix of questions for our critical items that probed for either the MC or the RCinterpretation. However, the relative clause probe is most relevant to evaluating the successful parsing and reanalysis of the garden path. By mixing the questions, syntactic adaptation in accuracy to the relative clause interpretation may have gotten washed out. Thus, rather than mixing the questions, we included a probe-type manipulation. This will allow us to better assess whether syntactic adaptation in reading times also has implications for the quality of the final interpretation, particularly if the quality of revision improves with exposure to RCs. If so, we would expect accuracy to the critical RC probe (vs. MCprobe) to demonstrate adaptation, like that observed for RTs. Likewise, this manipulation also allows us to further assess whether incongruent Stroop affects comprehension (i.e., in reaching the final correct parse) even if not observable in self-paced reading.

### 4.1. Methods

#### 4.1.1. Participants

A total of 229 native speakers of English (123 women, 105 men, 1 non-binary; Mage = 30.2, Sdage = 8.1) were recruited via the online recruitment platform Prolific (https://www.prolific.co) (accessed on 23 June 2021) following the same exclusion criteria as in previous experiments. Our internal check identified 9 participants who answered affirmatively to being colour-blind and one participant who stated that they were not native English speakers; all of these confirmed that this was selected by mistake when contacted.

#### 4.1.2. Materials

All the materials were identical to the ones used in Experiment 1, with the addition of a comprehension question probe manipulation. As the design was now 2 sentence ambiguity (ambiguous vs. unambiguous) by 2 Stroop (incongruent vs. congruent) by 2 question probes (MC vs. RC), we increased the number of experimental items from 36 to 40 to be divisible by 8. The 2 questions probed either the verb–argument structure of the relative clause, like 11a. and 12a., or the verb–argument structure of the MC, like 11b. and 12b. Additionally, 20 of the RCand 20 of the MC questions were also formulated in active voice, like 11a., 11b., and 12b., and the other half in passive voice, like 12a. Although some items

had both question types in either passive, active or a mix of the two voices, these were balanced over all items. All of the questions required a correct 'yes' response.

11.  The sunburned boys (who were) fed the hot dogs got a stomach ache.

    11a. Did someone feed the boys the hot dogs?—RC question in active voice.

    11b. Did the boys feel unwell?—MC question in active voice.

12.  The kitchen staff (who were) rushed in the cafeteria soon got very sleepy.

    12a. Were the staff rushed in the cafeteria?—RC question in passive voice.

    12b. Did the staff get drowsy?—MC question in active voice.

The addition of question probe type condition allowed us to investigate firstly whether the type of clause being probed (MC vs. RC) interacted differentially with accuracy, and secondly whether it interacted with the prior Stroop task. This also provided us with an opportunity to test whether the quantitative effects on reading times also result in a qualitative effect on interpretation as probed via accuracy.

The 72 fillers used in Experiments 1 and 2 were reduced to 68 in Experiment 3 for balancing purposes. The number of yes vs. no responses was also balanced across all items, as in the previous experiments.

### 4.1.3. Procedure

The procedure was largely the same as in Experiment 1, apart from the method of sentence presentation, where SPR was replaced by full-sentence presentation. The entire sentence appeared directly after a fixation cross. Here, the participants were instructed to press the space bar as soon as they were finished reading the sentence for the comprehension question to appear. As the constant pressing of the space bar provided sufficient engagement with the task, there was no need for a time-out limit for each individual word; however, reading a full sentence at once is a more passive task than SPR so we introduced a time-out limit as an additional motivation for the participants to keep on reading at a natural pace without disengaging from the task. We set this limit to 12 s to provide the participants with enough time to read and, hopefully, understand the sentences whilst also encouraging timely engagement. Once the time-out limit was reached, a question would automatically appear. The participants were instructed to read at their own pace but were warned that the sentences would eventually disappear. They were also explicitly told to try and press the space bar as soon as they finished reading before the time ran out. After excluding the participants based on their Stroop and filler performance, 2.88% of all sentence trials were timed out; otherwise, the mean reading time for all sentences was 3607 ms. This fairly low percentage of timed-out trials and reading times far below the allotted 12 s shows that participants were actively engaging with the task instead of passively waiting for the sentence to disappear.

### 4.1.4. Exclusions

We performed the same exclusion steps as before. A total of 53 participants were excluded as they were less than 80% accurate on filler questions and/or Stroop trials, or 23% of all participants recruited. Although this may seem concerning, as this is twice as many as in the previous experiments, there are two possible explanations. First, the participants who had participated in the previous 2 experiments could not participate in this one, this could have left us with a potentially less motivated pool of participants. Secondly, it is possible that SPR is a more active task, where full-sentence presentation is more likely to permit disengagement. Indeed, the 53 excluded participants had a time-out rate of 14.96% for filler trials compared to 2.61% for the remaining participants, indicating that they were more susceptible to distraction. Further, the remaining participants were on average 96% accurate on Stroop and 91% accurate in comprehending the filler sentences, comparable to the participants in the previous two experiments. Thus, despite having a higher exclusion rate, the exclusion criteria seem to be effective in eliminating poor performers.

As our sentences now had a time-out limit, the sentence exclusion criteria had to be changed from an absolute (i.e., any sentence with RTs > 20,000 ms) to a relative time limit. Thus, we excluded any trials with sentences with RTs that were more than 2.5 SD away from the average log RT for that specific item. This resulted in the exclusion of 0.14% or 26 of all trials.

### 4.1.5. Data Analysis

The same analysis steps as in Experiment 1 were performed, with the addition of a question-type fixed effect in the comprehension accuracy model. The residualisation procedure was the same as in the previous two experiments but accounted for the full length of a sentence in characters (excluding spaces) rather than each individual word.

### *4.2. Results*

### 4.2.1. Stroop Reaction Time

There was a standard Stroop effect ($\beta = -37.65$, SE = 11.84, df = 9.92, t = $-3.18$, $p = 0.010$) and a main effect of log(trial) ($\beta = -13.96$, SE = 1.49, df = 18,299.01, t = $-9.34$, $p < 0.001$), where RTs became significantly shorter as the experiment went on, replicating the results found in the previous two experiments.

### 4.2.2. Sentence RT

There was a standard effect of ambiguity for full-sentence RTs ($\beta = 0.07$, SE < 0.01, df = 38.24, t = 12.81, $p < 0.001$). Note that the unambiguous sentences have more words than the ambiguous ones, yet they still have a shorter reading time (4105 ms vs. 4307 ms). There were also main effects of experimental item trial ($\beta = -0.004$, SE < 0.01, df = 6966.75, t = $-5.11$, $p < 0.001$) and log(trial) ($\beta = -0.03$, SE < 0.01, df = 6933.89, t = $-3.36$, $p = 0.001$) where RTs progressively decreased as the experiment progressed.

There was also a significant ambiguity by experimental item trial interaction for the full-sentence RT ($\beta = -0.001$, SE < 0.01, df = 6931.19, t = $-3.26$, $p = 0.001$). Ambiguous sentence RTs decreased as the experiment progressed ($\beta = -0.005$, SE < 0.01, df = 6969, t = $-5.95$, $p < 0.001$) and unambiguous sentence RTs did as well but at a slower rate ($\beta = -0.003$, SE < 0.01, df = 6970, t = $-3.29$, $p = 0.001$) (Figure 8; note in the figure the RTs are slightly increasing for the unambiguous sentences but this is a limitation of plotting data that does not exactly represent the complex models).

Finally, there was also a main effect of Stroop ($\beta = -0.009$, SE < 0.01, df = 6761.74, t = $-2.41$, $p = 0.016$) not observed with SPR in Experiments 1 and 2, where all sentences (ambiguous and unambiguous) were read slower if they followed an incongruent Stroop rather than the congruent one. However, there was no Stroop by ambiguity interaction ($p = 0.956$; Figure 9).

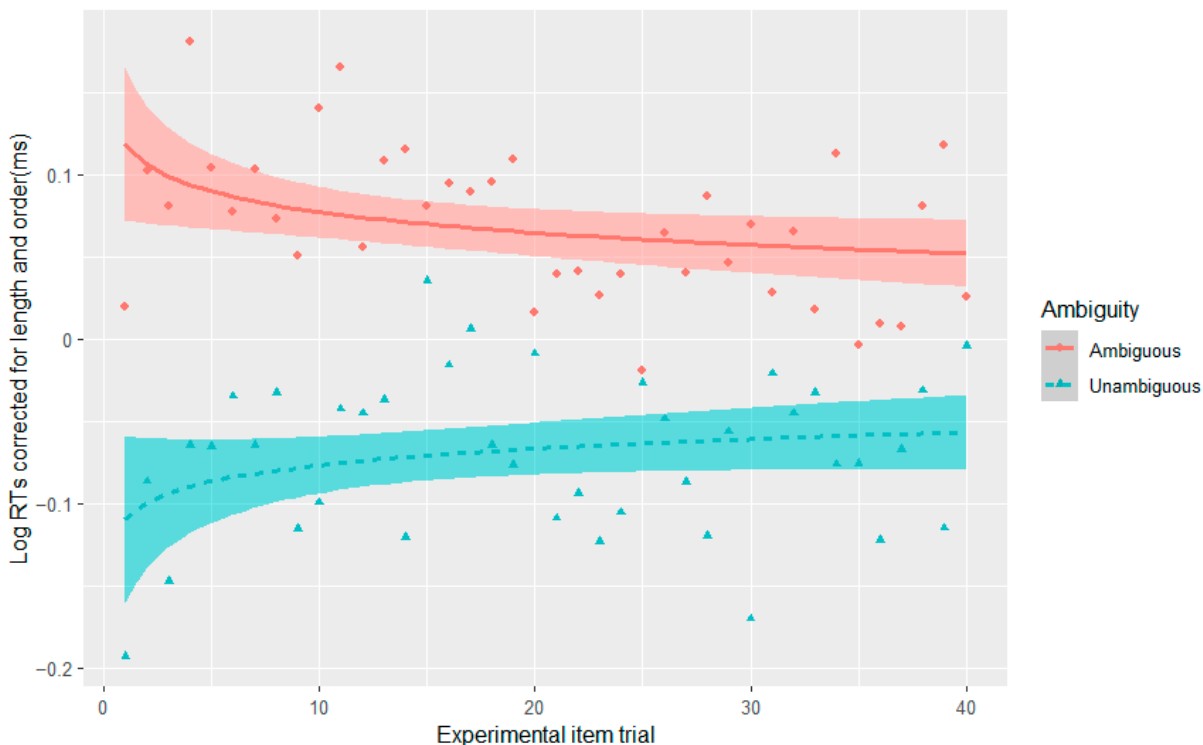

**Figure 8.** Change in Ambiguity Effect for Full-Sentence RT as a Function of Number of Experimental Items Seen for Experiment 3. Note that the plot demonstrates a decrease in the ambiguity effect (the difference in RRTs between ambiguous (red line) and unambiguous (green line) sentences). The trendline is presented log-linearly. Log RTs were residualised for word length and trial order to account for task adaptation to as best as possible approximate the model that included trial order as a fixed factor.

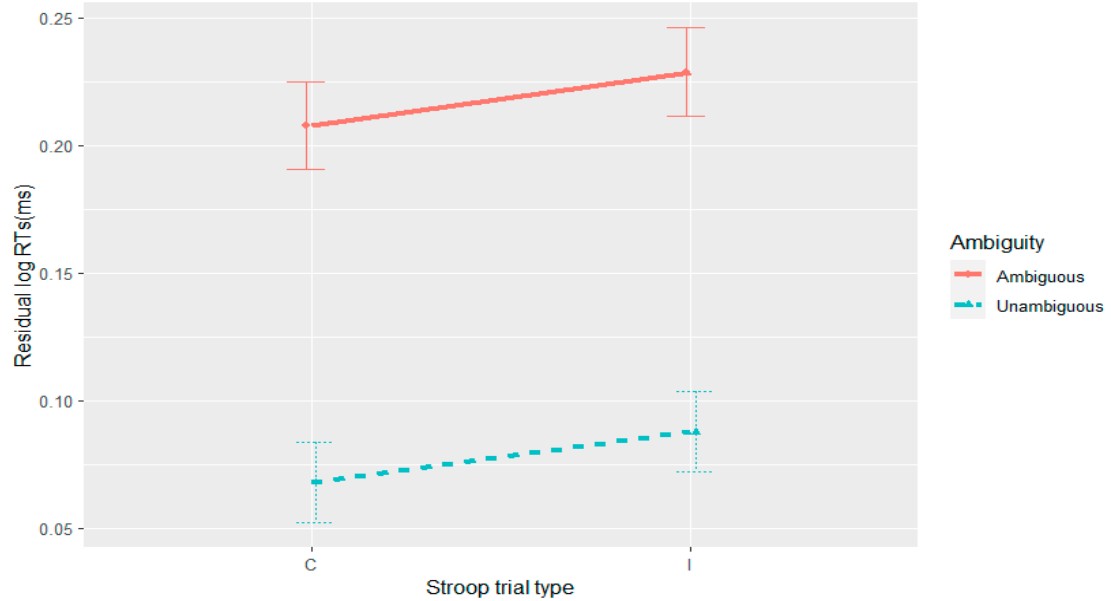

**Figure 9.** Ambiguity by Stroop Interaction at the Disambiguating Region in Experiment 3. Note that the plot shows main effects of Stroop, where all post-incongruent sentences took longer to read, and of ambiguity, where ambiguous sentences took longer to read than unambiguous sentences; however, the two effects do not interact. The figure illustrates mean residual log reading times with the error bars representing 95% confidence intervals.

### 4.2.3. Comprehension Question Accuracy

There was a standard effect of ambiguity (β = −0.30, SE = 0.06, z = −4.75, *p* < 0.001). There was also a main effect of question type (β = 0.28, SE = 0.13, z = 2.09, *p* = 0.037), where RC probes were less accurate than MC probes. There was also a main effect of the experiment item trial (β = 0.01, SE < 0.01, z = 2.69, *p* = 0.007), where accuracy generally increased as the experiment went on. However, the two effects did not interact (*p* = 0.480).

There was also a Stroop by ambiguity interaction (β = −0.11, SE = 0.04, z = −4.75, *p* < 0.001). Decomposing this interaction indicated that ambiguous sentence accuracy was higher post-congruent Stroop than post-incongruent Stroop (β = 0.218, SE = 0.11, z = 2.04, *p* < 0.041). The difference for unambiguous items was not significant (*p* = 0.057) but numerically went in the opposite pattern (Figure 10). This is inconsistent with conflict monitoring theory.

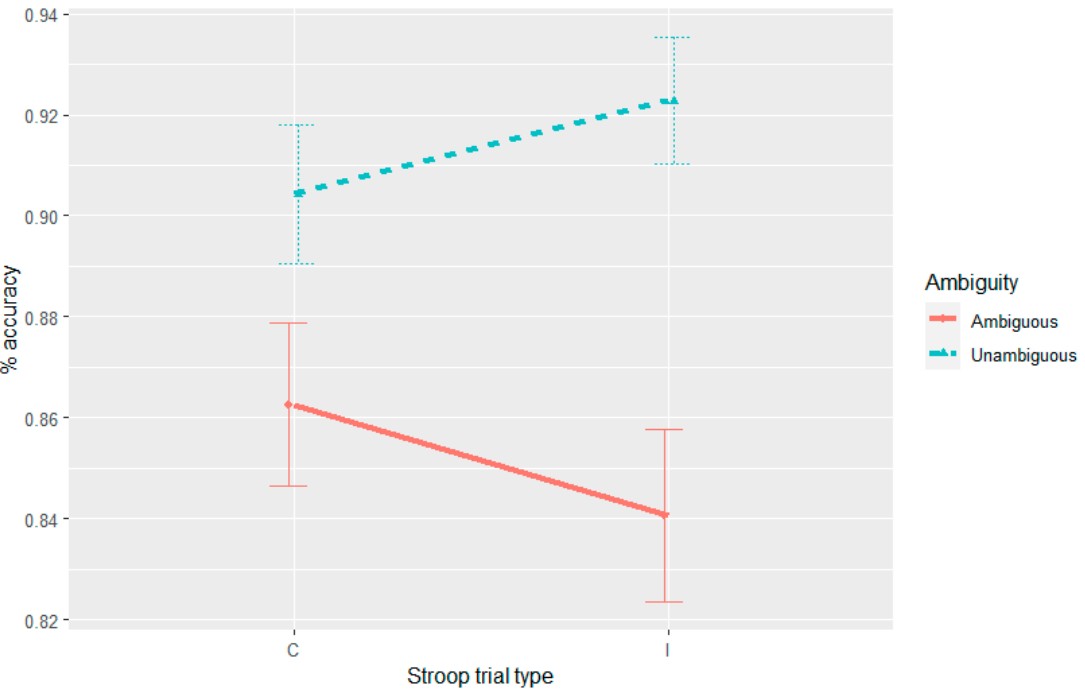

**Figure 10.** Ambiguity by Stroop Interaction for Accuracy in Experiment 3. Note that the plot shows a lower accuracy for ambiguous items preceded by incongruent Stroop than when preceded by congruent Stroop. There was no difference in accuracy on unambiguous probes. The figure illustrates percent accuracy with the error bars representing 95% confidence intervals.

Finally, there was also a significant three-way interaction between Stroop, question type, and experimental item trial (β = −0.009, SE < 0.01, z = 2.41, *p* = 0.016). Decomposing this interaction showed that the difference between MC and RC probe accuracy became smaller as the experiment progressed, but only if these followed incongruent Stroop trials (β = −0.029, SE = 0.01, z = −2.81, *p* = 0.005) but remained stable if they followed congruent trials (*p* = 0.562). This was due to post-incongruent RC probe accuracy increasing (β = 0.028, SE < 0.01, z = 4.09, *p* < 0.001) whilst MC probe accuracy remained unchanged (*p* = 0.846).

### 4.3. Discussion

Full-sentence presentation was successful in reducing the duration from Stroop conflict to the disambiguating region, considering the total sentence reading times for Experiments 1–3 (see Table 4). The exact time from Stroop offset to syntactic conflict cannot definitively be known, but in the best-case scenario the average reading time per sentence is reduced from Experiment 2 to 3 by a factor of 0.10, which allows us to approximate the average duration from the start of the sentence to disambiguation to be 3342 ms. Yet, we still do not

observe conflict adaptation. We also do not see the expected conflict adaptation effect in comprehension accuracy.

Nonetheless, we do see overall sentence reading times impacted by Stroop. Incongruent Stroop resulted in longer reading times regardless of ambiguity. This could be an example of Post-Conflict slowing (Forster and Cho 2014; Rey-Mermet and Meier 2017; Verguts et al. 2011). Interestingly, we do not observe this in self-paced reading. This ultimately adds to the complexity of interpreting how Stroop affects different tasks (Braem et al. 2014, 2019), but does not provide further information on the mechanisms that can facilitate garden-path processing, which is the goal of the current study.

We replicate syntactic adaptation for the full-sentence reading time. However, this is not observed in accuracy even with the probe manipulation. Thus, it seems that the improvement is within the mechanisms' efficiency and does not have an implication for the quality of the interpretations generated.

## 5. Experiment 4

Experiment 3 tried to minimise the duration from Stroop to disambiguation by using full-sentence presentation. In doing so, we successfully reduced the total sentence reading duration and by extension the duration from Stroop offset to sentence disambiguation. However, we still do not find evidence for conflict adaptation, despite, once again, observing syntactic adaptation. Another consideration for the null conflict adaptation effect is that our garden-path sentences had a more difficult reanalysis than those of Hsu and Novick (2016), which may have limited the effect that the cognitive control mechanisms could have. To ensure the absence of effect was not due to the choice of garden-path sentences, we used the same goal-modifier sentences as Hsu and Novick (2016) in Experiment 4.

### 5.1. Methods

5.1.1. Participants

A total of 135 native speakers of English (62 women (including transgender women), 73 men (including transgender men); Mage = 33.7, Sdage = 7.34) were recruited via the online recruitment platform Prolific (https://www.prolific.co) (accessed on 22 June 2022) following the same exclusion criteria as in the previous experiments. Our internal check identified 3 participants who answered affirmatively to being colour-blind and 3 participants who stated that they were not native English speakers; one person confirmed that they were not in fact a native speaker of English and were excluded from subsequent analysis.

5.1.2. Materials

We used the same 48 critical items used previously by Hsu and Novick (2016), like 13a. and 13b. In 13a., the ambiguity arises from the initial interpretation of the first PP *on the napkin* as the goal of *the frog*; this has to be re-analysed as its modifier at the second PP *onto the box*. This ambiguity is avoided by introducing the relativiser *that's* before the first PP in 13b.

  13a. Put the frog on the napkin onto the box.

  13b. Put the frog that's on the napkin onto the box.

We presented these sentences in SPR to exclude visual information bias. However, since these sentences are worded as instructions to manipulate objects in relation to other objects, we had to modify how we use them to ensure they make sense in a setting without any supporting visual information. During the instructions we provided sentence examples alongside a visual board like the one used by Hsu and Novick (2016) to suggest how the participants can represent the sentences in their minds. The instructions explicitly stated that they would not see these visual boards during the actual experiment. To mimic the offline action response measure, we used a forced-choice task with three types of questions that queried either the initial (14) or the final (15) positioning of the target, with the possible responses being the correct and the incorrect goal mentioned in the sentence, or queried

the object that was moved (16), with possible responses being the actual object that was moved, or another object mentioned in the sentence:

14. Put the frog on the napkin onto the box. Where was the frog initially (before being moved)?
- On the napkin/On the box

15. Put the bell on the napkin onto the pillow. Where is the bell now (after being moved)?
- On the napkin/On the pillow

16. Put the horse on the binder onto the scarf. Which object was moved?
- The horse/The binder

The two response options appeared on the left and on the right below the question; the corresponding 'left arrow' and 'right arrow' keys were used to select the correct response. The correct/incorrect responses were counterbalanced with their location on the screen. The three types of probes were not manipulated and were distributed equally among the 48 critical items.

In addition to the critical items, 48 filler sentences were also adapted from Hsu and Novick (2016). These sentences did not have any relative clauses and had only one PP that would always be interpreted as the goal of the initial PP. As these did not exclude any goal/modifier ambiguity, this prevented the participants from learning that the ambiguous PP is always a modifier. Each filler item was followed by one of two types of forced choice probes that either queried the final positioning of the moved object, or the object that was moved.

An equal number of experimental and filler trials meant we could not balance the Stroop–sentence and sentence–Stroop pairs across the experiment without creating a contingency where experimental sentences were more likely to appear after a Stroop trial. Thus, 24 of the filler trials followed a sentence–Stroop sequence, with an overall 72 trials following a Stroop–sentence sequence.

### 5.1.3. Procedure

The procedure was similar to the ones in the previous experiments but with modified instructions. As these sentences had never been previously used with SPR, the participants were also asked a question that tested their comprehension of the requirements of the task.

### 5.1.4. Exclusions

The same exclusions were performed as in the previous experiments. Thirteen participants who were less than 80% accurate on filler questions and/or Stroop trials were excluded, or 9% of all the participants recruited. Otherwise, the remaining participants were on average 93% accurate on Stroop trials and 95% accurate on filler questions, comparable to the participants in the previous two experiments.

We excluded 0.12% or 14 trials with sentences that took longer than 20,000 ms to read and a further 0.44% of all words that took less than 100 ms or more than 2000 ms to read.

### 5.1.5. Data Analysis

We performed the same analysis steps as in Experiment 1. We followed similar logic when dividing the goal/modifier sentences into regions as with the sentences used in Experiments 1 and 2: *subject*, [*relativiser*], *ambiguous region*, *disambiguating region*, *final word* (spill): (Put)/the frog/[that's]/on the napkin/disambiguating region onto the/final word box.

### *5.2. Results*
### 5.2.1. Stroop Reaction Time

There was a standard Stroop effect ($\beta = -28.96$, SE = 10.93, df = 9.91, t = $-2.65$, $p = 0.025$) and a main effect of log(trial) ($\beta = -26.9$, SE = 1.98, df = 11,227.57, t = $-13.59$, $p < 0.001$), where RTs became significantly shorter as the experiment went on.

### 5.2.2. Sentence RT

A summary of results for each of the three regions of interest is presented in Table 5.

**Table 5.** Coefficients and t-values for Residual log RTs for Each Predictor (Rows) and Each Sentence Region (Columns) in Experiment 4.

| Predictor | Ambiguous Region | | Disambiguating Region | | Spill (Final Word) | |
|---|---|---|---|---|---|---|
| | β | t | β | t | β | T |
| Ambiguity (=amb) | <0.01 | 1.20 | **0.02** | **5.07 \*\*\*** | **0.01** | **2.30 \*** |
| Stroop(=cong) | <0.01 | 0.49 | <0.01 | 0.49 | <−0.01 | −0.42 |
| Exp. Item trial | **<−0.01** | **−8.40 \*\*\*** | **<−0.01** | **−7.27 \*\*\*** | **<−0.01** | **−4.30 \*\*\*** |
| Ambiguity:Stroop | <−0.01 | −0.38 | <−0.01 | −0.20 | <−0.01 | −1.20 |
| Ambiguity:Exp. Item trial | <0.01 | 0.57 | <−0.01 | −0.08 | <−0.01 | −0.49 |
| Stroop:Exp. Item trial | <−0.01 | −0.06 | <0.01 | 0.99 | <−0.01 | −1.34 |
| Amb:Stroop:Exp. Item tr. | <−0.01 | −0.05 | <0.01 | 0.23 | <−0.01 | −0.22 |
| Log(trial) | **−0.04** | **−5.73 \*\*\*** | **−0.04** | **−5.34 \*\*\*** | **−0.07** | **−6.60 \*\*\*** |

Note. \* $p < 0.05$; \*\*\* $p < 0.001$. Bolded values indicate significant effects.

*Ambiguous Region.* There were main effects of experimental item trial (β = −0.004, SE < 0.01, df = 5748.16, t = −8.40, $p < 0.001$) and log(trial) (β = −0.04, SE < 0.01, df = 5758.70, t = −5.73, $p < 0.001$) where RTs progressively decreased as the experiment progressed.

However, unlike in the previous experiments, there was no main effect of ambiguity ($p = 0.236$).

*Disambiguating Region.* There was a standard ambiguity effect (β = 0.02, SE < 0.01, df = 120.63, t = 5.07, $p < 0.001$). There were also main effects of experimental item trial (β = −0.003, SE < 0.01, df = 5723.96, t = −7.27, $p < 0.001$) and log(trial) (β = −0.04, SE < 0.01, df = 5733.44, t = −5.34, $p < 0.001$), following the same trend as in the previous region.

There was no interaction between Stroop and ambiguity ($p = 0.842$) (Figure 11).

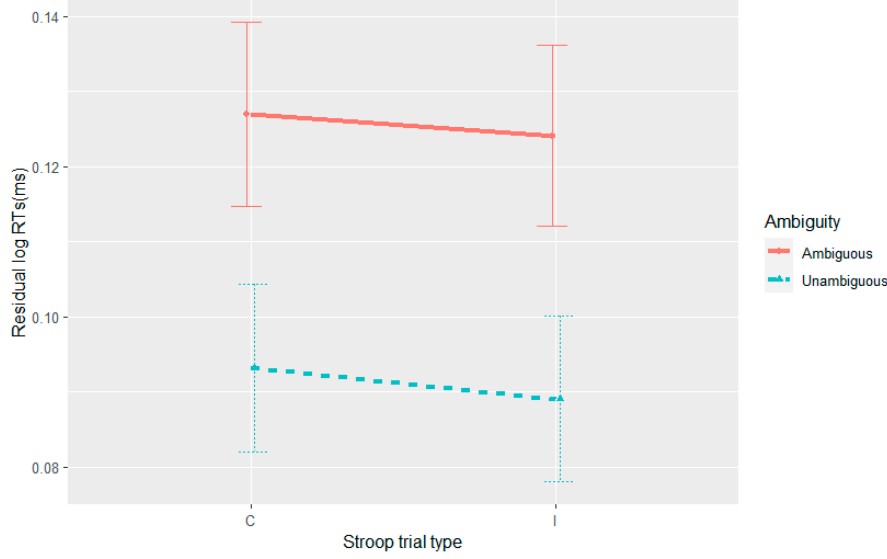

**Figure 11.** Ambiguity by Stroop Interaction at the Disambiguating Region in Experiment 4. Note. The plot shows a lack of interaction between Stroop and ambiguity effects. The figure illustrates mean residual log reading times with the error bars representing 95% confidence intervals.

### 5.2.3. Comprehension Question Accuracy

We found a standard ambiguity effect (β = −0.23, SE = 0.09, z = −2.68, $p = 0.007$). There was also an interaction between ambiguity and experimental item trial (β = 0.01, SE < 0.01, z = 3.44, $p < 0.001$) The simple main effects analysis showed that while accuracy

to ambiguous probes increased as the experiment progressed (β = 0.02, SE < 0.01, z = 3.80, *p* < 0.001), the accuracy to unambiguous probes remained stable (*p* = 0.216).

There was also an interaction between Stroop and experimental item trial (β = 0.009, SE < 0.01, z = 2.51, *p* = 0.012), with the simple main effect analysis showing that the average accuracy in post-congruent trials increased as the experiment progressed (β = 0.02, SE < 0.01, z = 2.86, *p* = 0.004), while the average accuracy on post-incongruent trials remained stable (*p* = 0.509).

Critically, these 2-way interactions with experimental items were tempered by a three-way interaction between Stroop, ambiguity and experimental item trial (β = −0.007, SE < 0.01, z = −1.97, *p* = 0.048). Accuracy to ambiguous probes increased regardless of whether it followed a congruent (β = 0.02, SE = 0.01, z = 2.93, *p* = 0.003) or incongruent Stroop condition (β = 0.02, SE = 0.01, z = 2.42, *p* = 0.016). For unambiguous probes, however, the accuracy decreased in post-incongruent trials (β = −0.02, SE = 0.01, z = −3.07, *p* = 0.002) but remained stable in post-congruent trials (*p* = 0.220; Figure 12).

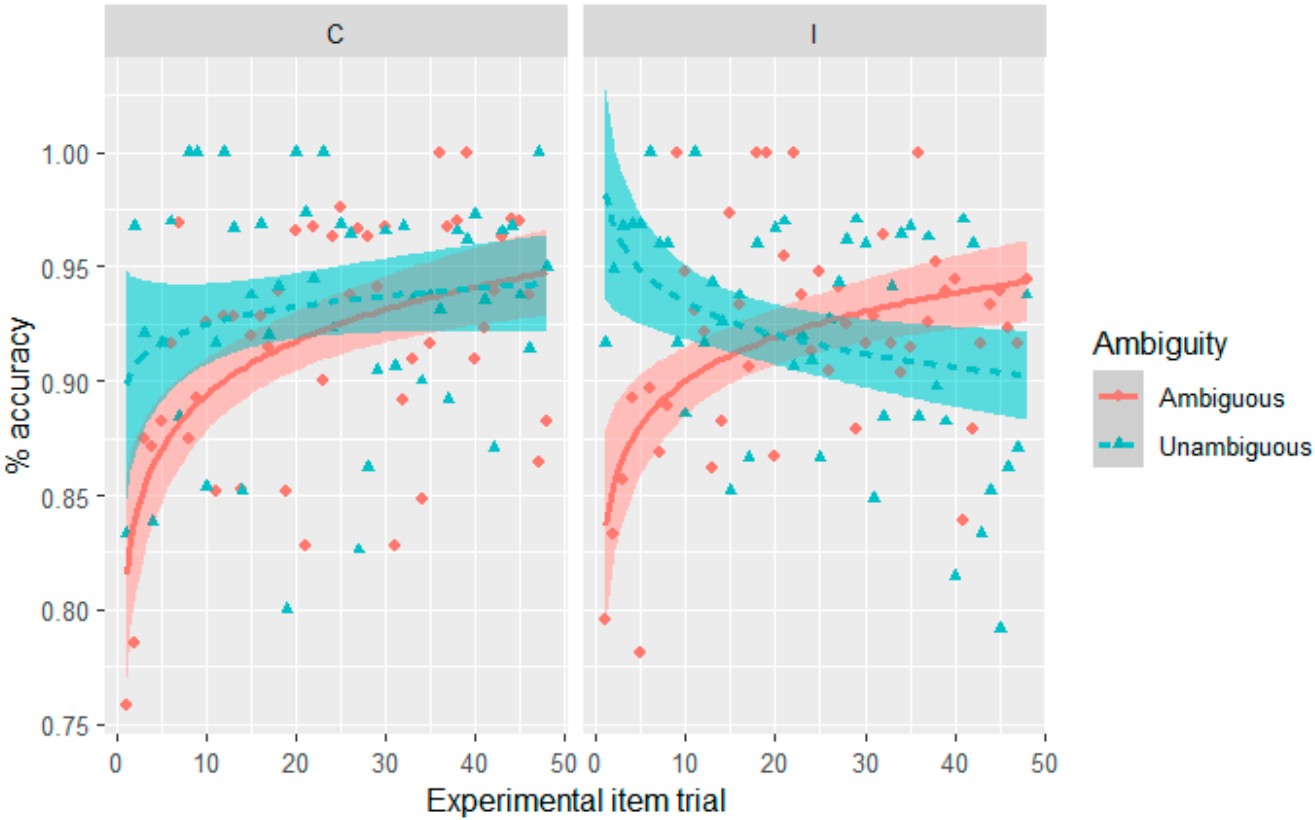

**Figure 12.** Interaction between Ambiguity, Experimental Item Trial and Stroop in Experiment 4. Note that the graph demonstrates a steady increase in accuracy for ambiguous sentence probes on both post-congruent and post-incongruent trials. Unambiguous sentence probes on the other hand decreased in accuracy, but only on post-incongruent trials, while remaining stable on post-congruent trials.

This pattern is similar to the three-way interaction in accuracy found in Experiment 1, in that a preceding incongruent Stroop task results in decreasing accuracy to unambiguous probes over the experiment. Again, this is not consistent with the predictions of conflict monitoring theory as the unambiguous sentences are devoid of conflict. Conflict monitoring expects the incongruent Stroop condition to engage conflict monitoring/resolution mechanisms to help in identifying and revising subsequent conflict. Nonetheless, the result is of interest in that incongruent Stroop is impacting syntactic adaptation in accuracy, but in a way not directly explicable by any current theory. This is something to be further explored by alternative theories.

### 5.2.4. Correlation between Stroop Effect and Ambiguity Effect

Given the lack of conflict adaptation across Experiments 1–4, we decided to look at a weaker claim for the role of cognitive control in ambiguity resolution by assessing the correlation between the participants' Stroop effect (based on the 72 or 81 (Experiment 2) Stroop practice trials) and ambiguity effect in RT in the disambiguating region. Ambiguity and Stroop effects were calculated as the difference between the average RTs for ambiguous and unambiguous conditions and incongruent and congruent conditions, respectively, for each participant.

This was not significant in any of the 4 studies (see Table 6). Not only that, but in all 4 experiments it demonstrated a numerically negative relationship. This means that the worse the individual's Stroop conflict resolution (bigger value for Stroop Effect), the better their ambiguity resolution (smaller value for Ambiguity Effect), the opposite of what conflict monitoring theory predicts.

**Table 6.** Descriptive Statistics and Correlations for Ambiguity and Stroop Effects for Experiments 1–4.

| Variable | *M* | *SD* | 1 | 2 |
|---|---|---|---|---|
| Experiment 1 | | | | |
|   1. Stroop | 72.47 | 50.42 | - | |
|   2. Ambiguity | 0.04 | 0.07 | −0.038 | - |
| Experiment 2 | | | | |
|   1. Stroop | 72.46 | 57.11 | - | |
|   2. Ambiguity | 0.05 | 0.08 | −0.006 | - |
| Experiment 3 | | | | |
|   1. Stroop | 71.57 | 54.50 | - | |
|   2. Ambiguity | 0.14 | 0.13 | −0.144 | - |
| Experiment 4 | | | | |
|   1. Stroop | 73.23 | 54.31 | - | |
|   2. Ambiguity | 0.04 | 0.08 | −0.034 | - |

### 5.3. Discussion

Using the same goal-modifier ambiguity as Hsu and Novick (2016) had the benefit of further reducing the duration from Stroop offset to sentence disambiguation to on average 3107 ms and still we did not observe conflict adaptation. Hsu and Novick (2016) also tried to avoid syntactic adaptation by providing the same temporary ambiguity in the fillers while disambiguating towards the more probable parse. As we also adopted this design feature it is not surprising we did not observe syntactic adaptation at the disambiguating region. This result provides evidence against an argument that conflict adaptation is less detectable due to the effect of syntactic adaptation. Another argument in support of this is the lack of a 3-way interaction between Stroop, ambiguity, and critical item trial in Experiments 1–3, which would be expected if conflict adaptation is reduced with syntactic adaptation.

Given that we were not able to observe a causal role of Stroop on syntactic disambiguation, we looked for a weaker relationship between the two. We calculated the correlation between conflict monitoring ability in non-syntactic conflict (Stroop effect based on the 72/81 practice trials) and syntactic conflict (ambiguity effect) in each of the 4 experiments, and none were significant. Thus, we fail to find either a causal or any relationship between cognitive control, in conflict-monitoring terms, and syntactic ambiguity resolution.

Interestingly, in the absence of syntactic adaptation in reading times, we did observe it in accuracy. However, this was further qualified by its interaction with Stroop, whereby the post-incongruent Stroop trials differed from the post-congruent Stroop trials in that unambiguous probe accuracy decreased over the experiment. This result is not directly explicable from the conflict monitoring perspective.

It is prudent to note, however, that the comprehension question format in this experiment differed from those in Experiments 1–3. Rather than being yes–no questions, it was a

forced choice task where they were asked about an object and provided with two objects to choose between. Given that we did not observe syntactic adaptation for sentence reading times, accuracy adaptation may not be related to any changes in ambiguity processing. Rather, it may reflect a task-learning effect where the format of the questions allowed the participants to learn something specific about the question format and strategize in completing them. The conditions under which adaptation can be observed in comprehension accuracy require further research.

## 6. Bayesian Model Analysis

We found no evidence for conflict adaptation from Stroop to garden-path conflict over four experiments. However, this result should be considered with caution, since a null effect does not signify the absence of an effect but rather the absence of evidence for an effect. For example, it is possible that the effect does exist but needs higher statistical power to be observed (Vasishth and Gelman 2021). The Bayesian framework on the other hand takes a different approach to hypothesis testing, as it directly tests the null hypothesis rather than testing the evidence against it. Bayesian models combine prior information with current data to obtain a probability distribution of plausible values for a given effect (i.e., the posterior distribution). Thus, it allows us to make more straightforward conclusions about the existence or absence of an effect.

Considering this, we applied Bayesian methods to the same linear mixed-effect models of RTs from the disambiguating region (Experiments 1, 2, and 4) or from the entire sentence (Experiment 3). This takes use of our prior knowledge by fixing a set of 'priors', or the expected values for each of the fixed and random factors, as well as the leftover random error (sigma). For this, we used the output from the frequentist models to set weakly to moderately informative priors at (0, 0.1) for all of the fixed and random effects as well as the sigma. We used a relatively high number of samples, with every model sampling four chains of 10,000 iterations each (run on four cores) with 250 warmup iterations on each chain. To ensure high validity of the posterior samples we also set the adapt delta parameter to 0.99 to decrease the number of divergent transitions and set the max tree depth to 15 (Bürkner 2017). The fixed-effects structure was identical to the frequentist models. We also were able to fit maximal random effect structure for both item and participant random effect, where ambiguity interacted with Stroop. We used the same contrasts for Stroop and ambiguity as previously. All of the analyses were run using the BRMS package (version 2.18.0) in R (Bürkner 2017).

Bayes factors (BFs) were calculated to quantify evidence for conflict adaptation as well as the interaction between conflict adaptation and exposure (the three-way interaction between Stroop, ambiguity and experimental item trial). A BF is a ratio that provides support for one model over another in how well they fit the data. We compared a full model (H1) to a model that did not contain the three-way interaction (H0), and separately to a model without the three-way interaction and the two-way Stroop by ambiguity interaction (H0). To calculate BF for conflict adaptation (the second H0 model), we had to exclude the full three-way interaction alongside the two-way Stroop by ambiguity interaction to avoid violating the principle of marginality (van Doorn et al. 2021; Wagenmakers et al. 2018). The Bayes factor was calculated using the bridge sampling method which provided the ratio of the marginal likelihoods of H1 and H0. BFs were interpreted according to Lee and Wagenmaker's scale (Lee and Wagenmakers 2014) where a BF over one indicates evidence for H1, and a BF under one evidence for H0. Specifically, a BF between 0.3 and 3 is seen as inconclusive (no evidence of either absence or presence of an effect); values between 3 and 10 provide moderate evidence, with values over 10 providing strong evidence in favour of H1; and values between 0.3 and 0.1 provide moderate evidence, with values below 0.1 providing strong evidence for H0. The BF was calculated a total of six times for each effect to check the stability of its calculation (Nicenboim et al. 2021).

*Results*

All Rhat values in the output for all four models were smaller than 1.1, indicating that our priors were sufficiently strong and/or the number of iterations was sufficient for the chains to converge (Bürkner 2017). Here, we will only discuss the results for the conflict adaptation effect as well as its moderation with exposure. The posterior for each effect within each experiment is summarised with its mean and 95% credible interval in Table 7.

**Table 7.** Posterior for each effect for each experiment is summarised as its mean and 95% credibility interval.

| | Experiment 1 | Experiment 2 | Experiment 3 | Experiment 4 |
|---|---|---|---|---|
| **Predictor** | *M* [95% CrI] | *M* [95% CrI] | *M* [95% CrI] | *M* [95% CrI] |
| Ambiguity (=amb) | 0.02 [0.01, 0.03] | 0.02 [0.02,0.03] | 0.07 [0.06, 0.08] | 0.02 [0.1, 0.3] |
| Stroop(=cong) | <0.01 [<−0.1, 0.01] | I−C:−0.01 [−0.02,0.01] N−I:0.01 [−0.01,0.02] | −0.01 [−0.02, <−0.01] | <0.01 [<−0.01, 0.01] |
| Exp. Item trial | <−0.01 [<−0.01, <−0.01] | <−0.01 [<−0.01, <−0.01] | −0.01 [−0.01, <−0.01] | <−0.01 [<−0.01, <−0.01] |
| Ambiguity:Stroop | <−0.01 [−0.01, 0.01] | I−C: < 0.01 [−0.01,0.02] N−I:0.01 [−0.01,0.02] | <−0.01 [−0.01, 0.01] | <−0.01 [−0.01, 0.01] |
| Ambiguity:Exp. Item trial | <−0.01 [<−0.01, <0.01] | <−0.01 [<−0.01, < −0.01] | <−0.01 [<−0.01, <−0.01] | <−0.01 [<−0.01, <0.01] |
| Stroop:Exp.Item trial | <0.01 [<−0.01, <0.01] | I−C: < 0.01 [<−0.01, < 0.01] N−I: < −0.01 [<−0.01, < 0.01] | <−0.01 [<−0.01, <0.01] | <0.01 [<−0.01, <0.01] |
| Amb:Strp:Exp. Item tr. | <0.01 [<−0.01, <0.01] | I−C: < −0.01 [<−0.01, < 0.01] N−I: < 0.01 [<−0.01, < 0.01] | <−0.01 [<−0.01, <0.01] | <0.01 [<−0.01, <0.01] |
| Log(trial) | −0.07 [−0.1, −0.04] | −0.06 [−0.07,−0.05] | −0.3 [−0.5, −0.1] | −0.04 [−0.06, 0.03] |

The posterior estimate of the mean for the ambiguity by Stroop interaction in Experiment 1 was −0.001 ms, 95% CrI [−0.008, 0.005] with a BF of 0.00009–0.00010 (the range for all six BF calculations), indicating strong evidence against conflict adaptation; in Experiment 2 it was 0.005 ms, 95% CrI [−0.006, 0.016] for the incongruent–congruent contrast and 0.005 ms, 95% CrI [-0.006, 0.017] for the neutral–incongruent contrast with a BF of <0.00001–< 0.00001, indicating strong evidence against conflict adaptation; in Experiment 3 it was <−0.001 ms, 95% CrI [−0.009, 0.009] with a BF of 0.00003–0.00058, indicating strong evidence against conflict adaptation; and in Experiment 4 it was −0.001 ms, 95% CrI [−0.007, 0.006] with a BF of 0.00005–0.00009, indicating strong evidence against conflict adaptation. Overall, the Bayes factors for all four experiments provide evidence against conflict adaptation.

The posterior estimate of the mean difference for the ambiguity by Stroop by experimental item trial interaction in Experiment 1 was <0.001 ms, 95% CrI [−0.001, 0.001] with a BF of 0.00284–0.00324, indicating strong evidence against conflict adaptation being moderated by exposure; in Experiment 2 it was <-0.001 ms, 95% CrI [−0.002, <0.001] for the incongruent–congruent contrast and <0.001 ms, 95% CrI [<−0.001, 0.002] for the neutral–incongruent contrast with a BF of <0.00001–< 0.00001, indicating strong evidence against conflict adaptation being moderated by exposure; in Experiment 3 it was <−0.001 ms, 95% CrI [−0.001, 0.001] with a BF of 0.00212–0.00358, indicating strong evidence against conflict adaptation being moderated by exposure; and in Experiment 4 it was <0.001 ms, 95% CrI [<−0.001, <0.001] with a BF of 0.00166–0.00267, indicating strong evidence against conflict adaptation being moderated by exposure. Overall, the Bayes factors for all four experiments provide evidence against conflict adaptation being moderated by exposure to experimental structures.

## 7. General Discussion

The current set of four experiments aimed to merge work on adaptation with garden-path processing over different time scales: trial and experiment. At the trial level, adaptation has been attributed to residual activation of a conflict monitoring mechanism from the preceding trial/stimulus (Hsu and Novick 2016). At the experiment level, it has largely been assumed to be due to implicit learning by (a) mechanism(s) operating over incremental exposure to garden-path sentences (Fine and Jaeger 2016; Fine et al. 2013; Prasad and Linzen 2021; Yan and Jaeger 2020). There has been debate as to what mechanisms can learn on

the longer timescale: the parser's assignment of probability to the temporarily ambiguous parses (Fine and Jaeger 2016; Fine et al. 2013), the parser's ability to revise the incorrect parse (Yan and Jaeger 2020), task-based learning (Prasad and Linzen 2021), or a domain-general cognitive control mechanism that assists with detecting conflict and conducting revision (Hsu et al. 2021; Hsu and Novick 2016; Sharer and Thothathiri 2020). The strength of a domain-general conflict monitoring mechanism comes from it being commensurate with facilitated garden-path processing, both from trial-to-trial and over longer periods of exposure. A limitation of this work is that the trial-to-trial findings depend on the visual world paradigm which does not necessarily directly measure syntactic processing. Thus, we used two reading paradigms that avoid an interaction with visual object processing—self-paced reading and standard reading under full-sentence presentation. If a domain-general conflict-based cognitive control mechanism is underlying both trial- and experiment-level adaptation effects, we predicted we would observe a reduction in the ambiguity effect at the disambiguating region when the preceding Stroop condition was incongruent, and we expected the ambiguity effect to decrease over the course of the experiment. We failed to find evidence of conflict adaptation in each of the four experiments, despite the manipulation of sentence ambiguity and Stroop congruency being successful, as evidenced by the standard Stroop and ambiguity effects. Likewise, we observed syntactic adaptation across Experiments 1–3 where it was predicted.

### 7.1. Conflict Adaptation

A null effect of conflict adaptation is difficult to argue for; however, there are a few arguments that seem relevant. First, these were high-powered experiments (Exp. 1, $n = 96$; Exp. 2, $n = 168$; Exp. 3, $n = 176$; Exp. 4, $n = 122$) relative to the original work demonstrating conflict adaptation in sentence processing ($n = 41$ in Kan et al. 2013; $n = 23$ in Hsu and Novick 2016; $n = 26$ in Hsu et al. 2021). Second, in Experiment 4, we used the same garden-path stimuli as Hsu and Novick (2016), as well as the same principles of design for the fillers, and did not observe the effect. The fourth experiment also did not result in syntactic adaptation in reading times given the design of the fillers, such that one cannot argue that conflict adaptation disappears or weakens with syntactic adaptation. This is further supported by the absence of a three-way interaction between Stroop, ambiguity, and experimental item trial in Experiments 1–3. If conflict adaptation were to weaken with syntactic adaptation, one would expect that conflict adaptation would decrease with increasing exposure to the less probable interpretation over the experiment.

Nonetheless, one could always argue that the duration from Stroop offset to disambiguation is longer in our reading experiments than in the sentence listening experiments of Hsu and Novick (2016; see Table 4), and that this could account for our failure to observe conflict adaptation. Indeed, on average the shortest duration across our experiments was 3107 ms, whereas in Hsu and Novick (2016) this was calculated to be 2625 ms (see Table 4). However, studies that have manipulated the ISI in Stroop-to-Stroop trial designs found the adaptation effect only disappeared in the 4000–5000 ms ISI condition (Egner et al. 2010). Thus, in principle, our duration is within the realms where conflict adaptation has previously been observed. Even if the Stroop-to-disambiguation duration was too long to observe trial-level adaptation in our experiments, we would still expect to observe a simple relationship between Stroop performance and ambiguity resolution. Specifically, we would expect that individuals with a more efficient conflict monitoring mechanism would have both smaller Stroop and ambiguity effects. However, we found no significant correlation between the size of the two effects across all four experiments. In fact, the numerical relationship between Stroop and ambiguity effects was negative, opposite to what was expected. Thus, we fail to provide either the stronger (i.e., conflict adaptation) or weaker evidence (correlation between Stroop and ambiguity effects) for a domain-general cognitive control, couched within the conflict monitoring account, to facilitate garden-path processing.

Finally, with inferential statistics, the absence of evidence does not provide evidence for the lack of an effect, which led us to run a Bayesian analysis to directly test the null hypothesis. Across all four experiments, the Bayes factor provided support for the null hypothesis, both for conflict adaptation and for conflict adaptation interacting with experimental item exposure.

This support for the null hypothesis does not imply that conflict adaptation cannot be found under alternative conditions. Indeed, in the introduction we raised the possibility that the observed conflict adaptation from incongruent Stroop to garden-path processing in prior work (Hsu et al. 2021; Hsu and Novick 2016) does not directly arise at the level of syntactic processing but its integration with the visual scene. Thus, it is happening at the level of visual object processing. If that is the case, then those prior results would not provide evidence for a domain-general mechanism per se, as the transfer would be happening from the visual conflict in Stroop to the visual object conflict created by the visual world set-up. Indeed, in Experiment 4 we did not conceptually replicate the conflict adaptation effect when using the same goal-modifier garden-path sentences in the absence of any accompanying visual information. If this argument is on the right track, it is still possible that a common mechanism is being recruited differentially across conflicts in different representational formats. Critically, its recruitment by conflict in one representational format would have no impact on a conflict in another format. Likewise, the ability to resolve conflict in one format would not have implications for its resolution in another. This would at least require some domain-specific constraints and not be compatible with a completely domain-general mechanism.

Future work should look at additional trial-to-trial adaptation effects to better establish when they do or do not occur. Can they occur when the preceding stimulus involves another syntactic conflict (regardless of the specific structure) or are more specific conditions required, such as the same temporarily ambiguous structure or the unambiguous version of the less preferred structure? The latter two cases are also examples of syntactic priming (Tooley and Bock 2014). Syntactic priming is the observation that a sentence is more likely to be produced with the same structure as the preceding sentence(s) or to be read faster when it has the same structure as the preceding sentence(s). In syntactic priming experiments, an initial sentence is presented (i.e., the prime) in one of several possible syntactic structures and a following sentence (i.e., the target) is either produced or comprehended by the participants. Most relevant to the current work is a recent study that found priming of the reduced relative structure in reading time measures, when the prime was a reduced relative clause compared to a MC (Tooley 2020). Priming has been interpreted as being due to either continued activation of the primed structure and/or error signalling, that is, the updating of mental syntactic models in response to a surprising sentence structure (up-regulating the surprising structure) (Chang et al. 2006; Tooley 2023). Rather than conflict-monitoring mechanisms remaining active from the prime, it is argued that activation of the syntactic structure of the prime is maintained, which facilitates subsequent processing of the same structure. An alternative account is error updating. It states that in the face of a temporary ambiguity and after receiving an error signal when selecting the incorrect parse an update to parse probabilities occurs, facilitating subsequent production or comprehension of that structure (Chang et al. 2006). In this way, the mechanism of syntactic priming would strengthen the structure's representation and could contribute to the cumulative effects of syntactic adaptation observed over an experiment. However, in Tooley's (Tooley 2020) recent study the prime (reduced relative clause) and target (reduced relative clause) of the key priming effect were alike both in terms of structure and having a syntactic conflict. Further investigation into different properties of the prime stimulus would shed light on whether trial-to-trial adaptation can occur across independent syntactic conflicts (i.e., syntactic conflict) or when using a consistent structure in the absence of conflict (i.e., pure syntactic priming). The latter could be achieved by using an unambiguous RC vs. MC as prime and an ambiguous reduced relative clause as target (this work is currently underway).

Although we did not observe the expected effect of the Stroop manipulation on sentence processing, in Experiment 3 we did see an effect of Stroop on overall sentence reading times, regardless of ambiguity. This might be consistent with post-incongruent slowing (Forster and Cho 2014; Rey-Mermet and Meier 2017; Verguts et al. 2011). Given this effect was not observed in the self-paced reading experiments, it reinforces the notion that Stroop effects are complex and can interact with the task at hand. As this is not directly affecting our main interest—conflict-monitoring mechanisms facilitating garden path processing—we will leave further interpretation aside.

There were also instances where Stroop affected comprehension accuracy, but these were not in line with the theoretical frameworks under discussion. Other than the ambiguity effect, the effects on accuracy are rather inconsistent across experiments. This seems to be in line with other work looking at comprehension accuracy that varies across individuals and tasks (Caplan et al. 2013) and may stem from the additional mechanisms that are recruited in answering comprehension questions, including strategies and memory (Meng and Bader 2021; Paolazzi et al. 2019).

### 7.2. Syntactic Adaptation

Experiments 1–3 were successful in replicating syntactic adaptation to the RC-MC ambiguity. A novel observation was its appearance under full-sentence reading without any equipment (i.e., for eye-tracking), a more natural condition. This again provides evidence against an account of syntactic adaptation that is based solely on learning the self-paced reading task and its interaction with sentence difficulty (Prasad and Linzen 2021). Failure to find conflict adaptation or a correlation between the Stroop and ambiguity effects also makes it unlikely that the syntactic adaptation effects are due to reanalysis improving via a domain-general cognitive control mechanism, as outlined by the conflict monitoring theory. At the very least, the evidence from the current experiments is not easily compatible with a higher-level reanalysis mechanism without some domain-specific constraints. Given that syntactic adaptation was localised to the disambiguating region and not observed in comprehension accuracy, the observed syntactic adaptation effects appear most compatible with either the parser changing expectations for the RC parse via probability updating or/and its reanalysis mechanisms improving in efficiency (Yan and Jaeger 2020).

In Experiment 4, we did not observe syntactic adaptation with the goal-modifier ambiguity in reading times. Given that there is a recurrent reanalysis in Experiment 4, this null result is at odds with a facilitated reanalysis account of the syntactic adaptation in reading times that were observed in Experiments 1–3. An expectation-based account can, however, provide an argument for the absence of syntactic adaptation in reading times in Experiment 4. Unlike Experiments 1–3, in Experiment 4 the ambiguity appeared in all the fillers and was always resolved to the more probable (goal) parse. This resulted in the goal-modifier ambiguity being resolved to a modifier interpretation 50% of the time and the goal interpretation the other 50%. While in Experiments 1–3 the RC-MC ambiguous verb was resolved towards the RC interpretation 100% of the time, other studies (Fine et al. 2013) have observed syntactic adaptation with the RC-MC ambiguity when the RC parse only occurred 50% of the time. As mentioned in the Introduction, a likely prerequisite for a detectable change in expectation for the less preferred parse is a sufficiently big change in its probability of occurrence in the natural environment to that in the experiment. Based on the verbs from Fine et al. (2013), the probability of an RC parse is only 0.008 in nature, such that a 0.5 occurrence in the experiment is a dramatic change (Fine et al. 2013). While our PP modifier can also be argued to be contained in a (reduced) relative clause, we would need to assume that the frequency of those RCs in nature is closer to 0.5 than those containing verbs that are ambiguous between the past participle/past tense. If so, the probability of a PP modifier interpretation in Experiment 4 may not have sufficiently differed from its occurrence in the natural environment to observe an impact on parsing expectations. In sum, it is possible that syntactic adaptation is not only dependent on the probability that the less preferred parse appears in the experiment but also on how much that probability

differs from the natural environment, which has been previously suggested (Fine and Jaeger 2016). To test this, future work should study whether the goal-modifier ambiguity can demonstrate syntactic adaptation when the proportion of ambiguous goal-modifier PPs that are disambiguated to a modifier PP is increased to 100%, the best-case scenario.

The difference in probability of a parse in nature vs. the experiment has likewise been used to explain why the penalty for the more frequent MC parse in the RC-MC ambiguity is difficult to replicate in syntactic adaptation studies (Yan and Jaeger 2020). The difference in probability is just much smaller for the MC parse (0.7 vs. 0.5) than it is for the RC parse (0.008 vs. 0.5). This raises a caveat to interpreting our syntactic adaptation effects from Experiments 1–3 as being commensurate with a change in expectation for the two parses. We did not assess adaptation away from the preferred parse (i.e., a penalty to the processing time of the MC interpretation), which would provide the strongest evidence for a change to parsing expectations, a finding that is inconsistently observed in previous work (Dempsey et al. 2020; Harrington Stack et al. 2018; Prasad and Linzen 2021; Yan and Jaeger 2020).

Further complicating the picture is the observation of syntactic adaptation in accuracy in Experiment 4, albeit one that was modulated by the Stroop manipulation where accuracy to the unambiguous sentences progressively decreased on post-incongruent trials. Thus, the nature of this three-way interaction is not consistent with any of the theories outlined here.

While the work here and elsewhere (Yan and Jaeger 2020) demonstrates syntactic adaptation is not specific to the self-paced reading task, it may be dependent on other aspects of the experimental environment. Other work on syntactic adaptation fails to show transfer effects from a more natural reading of paragraphs to an experimental single-sentence presentation (Atkinson 2016). Thus, a factor that may facilitate syntactic adaptation is the presentation of sentences in isolation, which may put an attentional emphasis on the syntactic structure. Further work is needed to understand exactly what is required for adaptation to occur and whether or when it can transfer to the wild.

### 7.3. Future Work

In future work, it would be worthwhile to look at what properties of a directly preceding stimulus or prime sentence can facilitate processing of a garden path sentence: an independent syntactic conflict or the target syntactic parse presented unambiguously? If conflict adaptation is not observed across conflicting syntactic representations (syntax-specific conflict monitoring mechanism), consideration of other accounts of cognitive control from conflict monitoring should be considered for their role in garden path processing.

Future work should also investigate the conditions under which syntactic adaptation is or is not observed to better isolate the mechanisms responsible. In particular, whether the goal-modifier ambiguity can demonstrate syntactic adaptation when the proportion of ambiguous PPs are disambiguated to a modifier is maximised to one. Given we saw increasing accuracy to ambiguous probes only in Experiment 4, future work should look at whether the type of comprehension question (yes/no vs. two-choice selection) is critical to this. Finally, work should explore whether the properties of the experimental environment—such as single-sentence presentation—are a prerequisite for syntactic adaptation to occur over the short time frame of an experiment.

### 7.4. Conclusions

Across four experiments, we found the standard Stroop and ambiguity effects in our Stroop-to-sentence reading trials. However, we failed to find evidence that processing conflict in incongruent Stroop facilitates subsequent processing of a garden-path sentence (i.e., conflict adaptation). This was despite the efforts to minimise the duration from Stroop to disambiguation and using the same garden-path sentences and Stroop to sentence design as Hsu and Novick (2016) but substituting the visual world paradigm for self-paced reading. This difference—the visual world paradigm—seems likely to be a key factor in the differential results. We suggest that in the visual world paradigm, Stroop could directly impact visual object processing rather than syntactic processing. We also failed to find

a simple correlation between an individual's ability to resolve Stroop conflict and their ability to resolve syntactic ambiguity. Thus, we do not find support for conflict-based domain-general cognitive control in these high-powered experiments.

We replicate the syntactic adaptation effect with the RC-MC ambiguity and under more natural full-sentence reading conditions. While the data fail to support a domain-general cognitive control mechanism being implicated in both conflict adaptation and syntactic adaptation, it is possible that a language-specific conflict monitoring mechanism is operating in both and is something that should be explored in future work.

**Supplementary Materials:** The following supporting information can be downloaded at: https://www.mdpi.com/article/10.3390/languages9030073/s1, File S1. Stimuli experiments 1–4, data files containing raw data for Experiments 1–4, R script containing code to replicate all of the analyses mentioned in the paper

**Author Contributions:** Conceptualisation, project administration, writing—review and editing, V.K. and A.S.; funding acquisition, V.K.; supervision, A.S.; design, data collection, analysis, interpretation, writing—original draft preparation, final approval of the article, V.K., F.C., K.C., J.C., X.Q., C.V., Y.Z., Z.X. and A.S. All authors have read and agreed to the published version of the manuscript.

**Funding:** This study was funded by the AHRC under grant number AH/R012679/1.

**Institutional Review Board Statement:** As the project involved non-invasive research with healthy participants, ethical approval was provided by the Linguistics Department Ethics Chair (approval number LING-2021-02-02, approved on 2 February 2021).

**Informed Consent Statement:** All of the participants gave their informed consent before starting the study.

**Data Availability Statement:** All of the sentence stimuli, data, and analysis code for this study are available on the APA's repository on the Open Science Framework (OSF) and can be accessed at https://osf.io/mzd3w/ (accessed on 14 January 2024).

**Conflicts of Interest:** The authors declare no conflicts of interest.

## Note

[1] These studies were published after the collection of data for most of the experiments reported here. Results from Experiment 1 were presented at the 34th CUNY Conference on Human Sentence Processing (Kuz et al. 2021).

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
