# Peer review of "Trial-Level and Contiguous Syntactic Adaptation: A Common Domain-General Mechanism at Play?"

_languages, doi:10.3390/languages9030073_

Round 1
Reviewer 1 Report
Comments and Suggestions for Authors
Summary: This manuscript reports four experiments that seek to replicate findings of a decreased garden-path sentence effect when sentences follow an incongruent Stroop trial. Previous observations of this effect have suggested it implicates a domain-general cognitive monitoring system contributes to syntactic ambiguity resolution and perhaps syntactic adaptation effects. The current set of experiments use reading time and accuracy measures rather than auditory language processing combined with a visual world display to rule out a visual conflict resolution mechanism as the source of this effect. Across four experiments, no interaction was observed that would imply improved garden-path effects follow incongruent Stroop trials. However, Stroop, ambiguity, and syntactic adaptation effects were observed. The authors suggest these findings do not support a domain-general cognitive control mechanism contributes to recovery from garden-path effects and/or syntactic adaptation effects.
General Impression: Overall, I found this set of experiments to be novel, theoretically interesting, and thorough. The design and analyses appeared to be carefully considered, well executed, and transparently explained. The Introduction did a very nice job explaining the relevant findings and motivation for the current study. The results were also concisely but thoroughly reported and unpacked, and the figures were very clear and appropriate. I also appreciated having the statistical output tables. I found the interpretation of the results to be fair and considered and not over-reaching for the most part. My main criticism of the current version of the manuscript is that the Discussion is not as well-written and accurate as I found the Introduction to be. Specifically, there are many mis-used and missing commas and awkward sentences. Also, the way syntactic priming is explained and related to syntactic adaptation is not completely accurate or appropriately sourced/cited. I have included specific comments, below, that point to these (and other) issues in the text. I believe this paper will be appropriate for publication with some strategic editing.
Specific Comments:
1. Page 1, line 38: “processing of garden paths is facilitated over time…” It would be more accurate to say “is facilitated with increased exposure” since that is what leads to the facilitation, not just the passing of time.
2. Page 10, paragraph starting on line 425: Why us different color sets for congruent and incongruent trials? Isn’t there a possibility that participants will notice/implicitly learn this?
3. Pages 17, 29, 35 (and maybe others): “triple interaction” should be “three-way interaction”
4. Page 19, line 682: “…activate a conflicting semantics…” should be “activate conflicting semantics”
5. Figure 5 doesn’t require a legend since there is an x-axis with the same information.
6. Page 26, line 853: 12 seconds for a single sentence (even a garden path) seems very long. Also, the fact that they pushed the button before this 12 seconds was up doesn’t necessarily mean they were comprehending the sentence; they could just push the button after one pass.
7. Page 27, lines 904-905: “…at a faster rate than the increase in unambiguous sentence RTs…” Looking at Figure 8, it seems that unambiguous RTs actually got numerically slower as trials increased, not faster at a slower rate.
8. Page 37, lines 1141-1144: “While at the trial level, adaptation can be seen as being due to residual activation of a mechanism from the pre-ceding trial/stimulus, at the experiment level, it is largely assumed to be due to implicit learning by (a) mechanism(s) operating over representations.” This sentence doesn’t make sense to me as it is currently worded and it lacks the necessary citations. Specifically, how do you have residual activation for a mechanism? Activation would be for a representation, not a mechanism. Also, the assertion that residual activation happens at the trial level and this leads to implicit learning over many trials doesn’t make sense in terms of a residual activation or implicit learning account of structural priming/syntactic adaptation effects.
9. Page 37, line 1168: “The 4th Experiment also did not observe…” Experiments can’t observe something. Change the verb (observe) to “yield” or “result in” so that it makes sense with an inanimate subject.
10. Page 38, lines 1187-1190: “However, …moreover, the numerical direction of the relationship was even opposite to that expected.” This sounds awkward to me.
11. Page 38, lines 1212-1220: This is not an accurate description of syntactic priming. Syntactic priming effects (that occur trial to trial) have been observed in RRC processing using reading time/comprehension measures like the ones employed in this study (see Tooley, 2023 for a recent review). In fact, structural adaptation is theoretically assumed to be syntactic priming that accumulates over more exposures and time (e.g., Chang et al, 2006, 2012; Fine & Jaeger, 2013, Tooley & Traxler, 2018). Please revise for clarity and cite where appropriate.
12. Page 39, paragraph beginning on line 1232: It seems strange to begin a new paragraph here with “Likewise” that seems so connected to the previous paragraph.
13. Page 39, line 1250: “…they did not assess adaptation to the preferred parse…” Is this supposed to say “away from” the preferred parse? I can’t seem to follow the logic laid out in this sentence.
14. Page 40, lines 1287-1290: “Thus, syntactic adaptation…in line with what has been previously suggested (Fine & Jaeger, 2016).” The logic here seems false to me. While your results could be consistent with this idea, since you didn’t not manipulate this and cannot directly assess it in your data, you should soften your language here.
15. Page 41, lines 1334-1337: “While the data fail to…should be explored in future work.” There have been many studies on syntactic/structural priming/adaptation and not all of them involve structures with conflict that needs to be resolved. For example, something like an experiement-induced bias towards one alternative in a dative structure has been shown to persist with mass exposure (i.e., syntactic adaptation; see work by Kaschak and colleagues). How could a language-specific conflict monitoring mechanism explain those sorts of findings?
Comments on the Quality of English LanguageThe Discussion section has some awkward phrasings and many oddly placed and/or missing commas. Editing of this section is highly encouraged.
Reviewer 2 Report
Comments and Suggestions for Authors
In this study, the authors test for both conflict adaptation and syntactic adaptation by using a stroop test followed by ambiguous/unambiguous versions of reduced relative garden path sentences respectively. They find across four studies no evidence for conflict adaptation while consistently finding evidence for syntactic adaptation.
The theory behind this study, as well as its design, is well-thought out and will make a nice addition to this literature; however, the biggest issue I have is that the authors are leaning too heavily on null effects here. I'm not against publishing null findings, but we have to be very clear about what they mean. So, my only *major* revision would be to 1) clarify the power here other than just stating that it's high (it actually might be pretty low for a small interaction effect, see Harrington-Stack et al., 2018), 2) state that the lack of evidence in its strongest form still doesn't rule out the possibillity, and 3) perhaps, if it's within the authors' abilities, either conduct an actual power analysis or perform a Bayesian analysis of the interaction to get Bayes Factor evidence in favor of the null hypothesis so that absence of evidence might actually constitute evidence of absence.
Some minor notes too:
Rather than residualizing, which is introducing an additional layer of error, you could just enter in word length as a fixed effect for your models.
When discussing low power, you should really cite Harrington stack et al 2018.
"After decomposing the interaction, we found that the interaction was not from the expected change in accuracy to the ambiguous sentence probe, as it stays constant on post-congruent (p =.833) and post-incongruent (p= .368) trials, but accuracy decreases for post-incongruent unambiguous probes (β= - 0.03, SE= 0.01, z = -2.26, p= .024), with accuracy to the unambiguous sentences on post-congruent Stroop remaining stable (p= .059).
^ I feel like this should be discussed a bit at some point - one interpretation could be that conflict monitoring can lead to poor performance on unambiguous sentences, right?
"Simple main effects analysis showed that the ambiguity effect decreased over the experiment but only on post-incongruent Stroop trials (β= 0.04, SE= 0.01, z = 3.89, p< .001), and not post-congruent Stroop trials (p= .309). Accuracy on post-incongruent unambiguous probes progressively decreased (β= -0.02, SE< 0.01, z = -3.07, p= .002) but remained stable for the ambiguous probes (p= .687). This pattern exactly replicated the triple interaction for comprehension probes we found in Experiment 1 but is not consistent with the predictions of conflict monitoring theory"
^ I don't get why this isn't exactly what's predicted by conflict monitoring theory. It seems that if accuracy only improves following incongruent stroop trials then it's necessary for it to occur on a trial-level basis in order for syntactic adaptation to occur no? This needs to be explained more carefully.
I'm not sure what section 5.2.4 adds, I think it can be removed
"First these were high-powered experiments (Exp. 1, n=96; 1165
Exp. 2, n=168; Exp. 3, n=176; Exp.4, n=122)."
^ actually, see Harrington stack et al 2018, they're not that highly powered for three-way interactions...
In the discussion, Dempsey, Liu, & Christianson 2023 seems potentially relevant - they find offline evidence in favor of changing expectations, but do not find online evidence, suggesting that explicit knowledge of structural frequencies may be quicker to change than real-time expectations.
